# Associative linking for collaborative thinking: Self-organization of content in online Q&A communities via user-generated links

Noa Sher[1]*, Sheizaf Rafaeli[1,2]

**1** Department of Information and Knowledge Management, University of Haifa, Haifa, Israel, **2** Shenkar College of Engineering, Design, and Art, Ramat Gan, Israel

* nsher01@campus.haifa.ac.il

**Data Availability Statement:** The data underlying the results presented in the study are available at https://osf.io/9xvnm Originally downloaded from the Stack Exchange Data Dump at: https://archive.org/details/stackexchange.

## Abstract

Virtual collaborative Q&A communities generate shared knowledge through the interaction of people and content. This knowledge is often fragmented, and its value as a collective, collaboratively formed product, is largely overlooked. Inspired by work on individual mental semantic networks, the current study explores the networks formed by user-added associative links as reflecting an aspect of self-organization within the communities' collaborative knowledge sharing. Using eight Q&A topic-centered discussions from the Stack Exchange platform, it investigated how associative links form internal structures within the networks. Network analysis tools were used to derive topological indicator metrics of complex structures from associatively-linked networks. Similar metrics extracted from 1000 simulated randomly linked networks of comparable sizes and growth patterns were used to generate estimated sampling distributions through bootstrap resampling, and 99% confidence intervals were constructed for each metric. The discussion-network indicators were compared against these. Results showed that participant-added associative links largely led to networks that were more clustered, integrated, and included posts with more connections than those that would be expected in random networks of similar size and growth pattern. The differences were observed to increase over time. Also, the largest connected subgraphs within the discussion networks were found to be modular. Limited qualitative observations have also pointed to the impacts of external content-related events on the network structures. The findings strengthen the notion that the networks emerging from associative link sharing resemble other information networks that are characterized by internal structures suggesting self-organization, laying the ground for further exploration of collaborative linking as a form of collective knowledge organization. It underscores the importance of recognizing and leveraging this latent mechanism in both theory and practice.

## Introduction

The plethora of environments promoting knowledge sharing, accumulation, and transformation are described by the umbrella term "virtual knowledge collaborations" [1, 2]. Platforms for virtual knowledge collaboration act as a medium for transferring knowledge or ideas from

**Funding:** The author(s) received no specific funding for this work.

**Competing interests:** The authors have declared that no competing interests exist.

the minds of individuals to the collective level, via the sharing of digital artifacts [3, 4]. The amount of knowledge and thought shared within these communities is immense, and they hold great potential for the generation of integrated knowledge [5–7]. Virtual knowledge collaborations span from small-group task-focused settings such as academic courses to large-scale persistent discussions comprising thousands of participants. This work focuses on the prevalent and fast-growing phenomenon of large-scale collaborative knowledge-sharing discussions, a form of persistent conversation, aimed at sharing, distributing, and creating knowledge. These have become an inherent component of knowledge creation and innovation in the educational, academic, professional, and civic arenas [8–11].

A distinction can be made between two kinds of virtual knowledge collaboration environments. On the one hand, environments are explicitly aimed at producing shared integrated artifacts, like Wikipedia or collaborative academic virtual environments in which students produce a group product. On the other hand, discussion-centered environments revolve around sharing knowledge through open discussions or in Q&A formats, that do not strive to create a unified product [12]. Despite this, participants discussion centered environments have been found to actively engage in forms of content integration, and this appears as a substantial component of the knowledge contribution behavior [1]. In collaborative constructions of unified products, participants explicitly negotiate as a part of the collaborative knowledge construction process either through reiterated modifications of the product or through interaction with each other [13, 14]. In open discussion or Q&A settings, however, the digitally mediated discourse itself becomes the substrate for the collaborative formation of new meanings [15] and the co-creation of knowledge is reflected in the interaction between the contributors and in the mutual integration of content posted by others [1]. As these discussion-centered environments lack top-down organization and integration of knowledge, emergent organizing mechanisms based on collective participant activity are an instrumental part of the collaborative process [4, 16–18]. They are also one of its notable merits, as they allow for novel combinations to appear and so enable collective creativity [17, 19–21]. Establishing the associatively-linked discussions' network qualities is a necessary milestone in the road toward identifying and studying mechanisms for user-based content organization within the complex environment of multi-participant discussions. The current study aimed to demonstrate this point by presenting the structures formed within Q&A discussions' linked networks that emerge as a result of the users' practice of incorporating associative links within their responses. From a topological perspective, these links can transform the otherwise fragmented topology of the discussion into a networked one, enabling the emergence of complex organizing structures such as connected sub-networks, clustering, and densely connected modules [22, 23]. In an individual cognitive setting, the emergence of macro-level structures based on local semantic associations can be considered an indication of self-organization [24]. The paper builds on this notion and explores the network formations within the collective content formed through collaborative associative linking of Q&A-based virtual knowledge collaborations, and lays the ground for further study of these networks. This could add to cumulative research regarding the self-organizing aspect of such collaborations, which has mainly concentrated on the underlying social networks, as well as other forms of knowledge organization such as tagging networks. In that, it illuminates another layer of the interactivity within large-scale virtual knowledge collaborations.

## Theory and conceptual framework

### Self-organization as a component of knowledge-sharing behavior in social Q&A environments

Q&A community participants post individually, rather than work together to create a uniform collective output such as a wiki page [12]. Nevertheless, participants in these discussions

continuously interact with each other through the content they contribute, and integrate knowledge or ideas from others' content into their own [1]. In their most basic format, social Q&As consist of a collection of distinct questions that can receive multiple answers, typically revolving around a topic that defines the discussion's ontological environment. The discussion network comprises question posts, out of which answers branch out, sometimes topped with an additional comments layer. Q&A platforms often form interactive communities: participants interact with each other and with the content at several levels in an ongoing process [25]. While the participants' straightforward engagement with the discussion comprises either posting questions or answering others' questions, cumulative research suggests that they are also engaged in structuring and organizing the discussion content for the good of the community. Organizing efforts can include explicit activities of introducing shared meanings such as categorizing [25], or tagging [4, 17, 26, 27]. They can also include social activities that contribute to the discussion's shaping by enhancing the salience of certain questions or answers, e.g. voting or rating [28]. The current study turns the spotlight on another type of content-organizing activity in Q&A discussions which we refer to as cross-linking: the connecting of knowledge units of question-answers (-comments) sets within a discussion through cross-referencing links between posts by embedding URLs of other questions within the answers [4, 22]. These are applied by participants to share knowledge and insight regarding relations between different knowledge units and to point out other relevant content within the discussion [16, 29]. Cross-links improve the navigability of the discussion by creating meaningful paths for the participants to follow and by directing them toward relevant knowledge. In that, their application represents another layer of interaction with content that goes beyond replying, upvoting, and other more recognized interactions: the sharing of associations. This is consistent with a more contemporary view of the participants, not only as content providers but also as path designers who facilitate navigation routes among the ocean of content, for the benefit of their peers [30]. Cross-linking can be regarded as an organic bottom-up means for organizing shared knowledge by discussion participants: as opposed to content networks produced based on textual similarities, the cross-linked network reflects intentional meaning-making and taps into an important portion of the group's knowledge, which is the explicit knowledge about the associative, logical or causal connections between the units which compose it [31]. Accordingly, cross-linking can be viewed as a higher-order element of interaction with content, one that impacts the structure of the collective knowledge space and contributes to its organization for the community's benefit [16–18, 22]. While Q&A discussions do not produce an explicit collective product but are rather a collection of individual contributions, this work aims to demonstrate that by linking related knowledge units together, the community molds this otherwise fragmented collection into an integrative shared product that can be perceived holistically as a network.

From a topological perspective, cross-links can transform the topology of the discussion from a collection of detached tree-shaped formations of questions-answers-comments into a networked one, comprising a "network of networks". While the sub-network composing each knowledge unit has a fully hierarchical topology ("tree-shape"), the overarching "network of networks" is not hierarchical, and so it enables the emergence of complex organizing structures typically found in other information networks such as citation networks or social-reaction networks [32]. These include large connected sub-networks, local clustering, and densely connected modules [21, 22].

Fig 1 depicts a discussion graph shaped by cross-links, compared to a threaded hierarchical discussion comprising only sequential links.

To demonstrate the effects of the links and highlight their role in shaping Q&A discussions' network topologies, we 1) compare indicators of organizing structures in the networks of eight

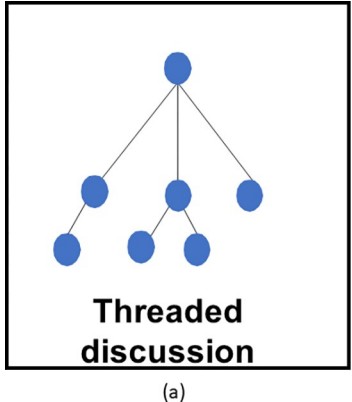 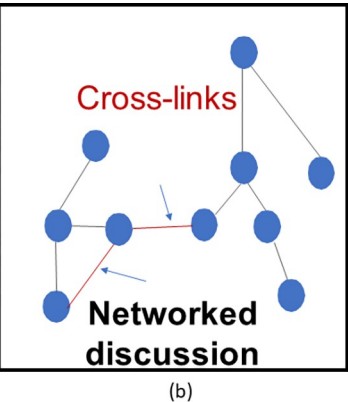

**Fig 1.** A schematic graph of a threaded hierarchical discussion (a) vs.a cross-linked discussion (b).

different Q&A discussions from the Stack Exchange platform, with the distribution of comparable indicators extracted from simulated null models with random links replacing the original links that were produced by the participants and 2) review how the results of these comparisons evolve over time.

The current study adds to existing work in several ways. First, it addresses large-scale discussions each consisting of thousands of posts over several years, occurring within real-world settings. Previous research on cross-link networks has either been based on academic environment small-scale discussions [21, 29] or excerpts from large networks, concentrating on the networks surrounding specific knowledge units [22]. Studies addressing entire discussions or larger portions of discussions have explored other types of edges such as shared tagging [4, 17, 27], social interactions [28, 33, 34], or textual similarities [35]. Second, using the distribution of indicators extracted from comparable null models with random links of identical size has allowed us to examine the cross-links' unique role in shaping the topology of the knowledge network and so establishing them as a form of self-organization that relies on user knowledge-sharing behavior. Third, reviewing several different discussions allowed for the exploration of this phenomenon across different networks [27], produced by different communities, and so demonstrating differences between discussions in the topology. Fourth, adding a timeline to the analyses offers insight into the trends that this process may follow, including the relations between evolving network topologies and external events that influence the content of the discourse through the participants' newly formed associations. This view of cross-linked Q&A discussions also has practical implications, as it can be used for extracting shared meanings formed by explicitly added connections.

## The topological indicators of emergent organizing network structures

Within Q&A communities, "knowledge units" comprising questions and answers connected through cross-links form networks that can be studied using a network analysis approach [22]. By extracting network metrics, we attempted to demonstrate the emergence of topological organizing structures that would indicate a collaborative structuring of the collective knowledge artifact. The network metrics were used for evaluating network properties and for discovering emergent structures, by comparing them with equivalent metrics that were extracted from null models in which random links replaced the cross-links (see Methods and Materials section for more details).

The properties that were investigated were:

1. **Defragmentation** of the networks. Cross-links can create connected subgraphs of knowledge units that are viewed by participants as conceptually related ("connected components"). They can also bridge across more loosely related content through various associations and connect subgraphs to an overarching component. This can increase the integration and reduce the fragmentation of the discussion network. The indicators that were used for assessing defragmentation were the size of the largest sub-network of connected units ("giant component") in comparison to a randomized null model, and the prevalence of large connected components comprising ten knowledge units or more.

2. The rise of local **focal points**. While most units have little or no connections to other units, some of the knowledge units become notably more connected than others and emerge as local focal points of the discussion. The indicators for the emergence of local focal points were the number of units with a degree of at least 4 (i.e., connected to at least four other knowledge units) and the maximal number of units connected to a single unit—both compared to equivalent metrics in a randomized null-model. Units that are associated with multiple other units can be regarded as local focal points. The distribution of the indicators extracted from the simulated random networks of similar density was used to verify whether these focal points are an artifact of link distribution and would be typically found in random networks with a similar edge density or rather a trait of the associative-link networks. The emergence of local focal points within the collective product of the discussion represents an emergent hierarchy in which some knowledge units are more central than others, allowing for more informed navigation of the knowledge space and facilitating the extraction of prominent themes.

3. **Micro-level organization**. Within network organizations of content, such as the individual semantic lexicon or Wikipedia, concepts that are associated with each other often share other associations as well [36, 37]. This is a form of micro-level organization because it results in easier navigation between closely related concepts. In network terms, this is expressed in increased triadic closure, which means that if two units are connected, and one of them is connected to a third unit, there is an increased probability that the other will also be connected to that unit. This is indicated by a higher clustering coefficient, compared to that of a randomly connected network [38]. The absolute number of triangles was calculated as an additional indicator as the clustering coefficient which is more commonly used is dependent on the potential number of triangles.

4. **Modular Formation.** Many real-world networks are modular in structure: smaller clusters of nodes are more densely connected amongst themselves in comparison to their connections to the network as a whole [39]. This creates a complex hierarchical organization of the network [38, 40] and formulates mesoscopic structures within the network that carry important information at an intermediate scale [41]. Relatively high modularity within the connected subgraphs would thus indicate another level of emergent organization, as well as replicate and elaborate on findings from Ye et al. that demonstrated high modularity within a subgraph centered around a "seed" node within the StackOverflow network [22]. Notably, within the individual human mind, a modular organization of the semantic lexicon, or knowledge of the world, is considered more adaptive in terms of accessing relevant information [36]. If the cross-links increase modularity, this could be another indication that the participants' linking activity assists in facilitating effective navigation for other community members.

Table 1 summarizes the network properties and their corresponding metrics:

**Table 1. Operationalization of network properties.**

| property | operationalization within the network | Graph metrics |
|---|---|---|
| Defragmentation and convergence | Forming of large connected components | • Size of largest connected components<br>• % of nodes in components>10 |
| The emergence of local focal points | Knowledge units with multiple connections | • % of nodes with degree$\geq$4<br>• Max degree |
| Micro-level organization | Clustering | • Global clustering coefficient<br>• The absolute number of fully connected triplets of nodes (triangles) |
| Mezzo-level organization | Module formation: distinct areas of denser connectivity within the connected subgraphs | • Modularity index |

Notably, in the associatively linked knowledge unit networks, a relatively small number of units became connected at all. The share of these varies between different communities (see Materials and Methods section for details). This characteristic that seems inherent to the domain has implications for network density and hence for network indicators that are related to network density, such as the degree distribution and the share of the largest connected component. The benchmarks that were used here to indicate network properties were created based on the characteristics of the networks within the sample that was examined. Consequently, comparisons were made to simulated randomly linked networks with similar link densities (see Materials and Methods section for more details).

Another important point is that while, as explained above, each discussion can be perceived as a "network of networks" with the knowledge units of question-answers-comments as subnetworks, the indicators were extracted for the overarching network consisting of whole knowledge units rather than for the full network containing the answers and comments branch-outs. While this simplification of the network loses some of the information, such as the number of each question's answers and comments indicating its collective perceived significance, it does not affect the topology of the cross-linked network and its emergent structures, which were at the focus of the current work.

## Research hypotheses

**Based on these, we hypothesized that.** H1: Metrics representing a complex internal structures such as the percentage of knowledge units with multiple connections, the global clustering coefficient and absolute number of triangles, the size of the largest connected components, and the prevalence of large connected components of the networks will be increased compared to the same metrics produced based on corresponding null-models with random links. To test this, distributions of the metrics extracted from 1000 iterations of simulated randomly linked networks were produced and the indicators of the real networks were compared to confidence intervals formed on the basis of corresponding estimated sampling distributions that were created using bootstrap resampling based on the metrics drawn from these iterations.

H2: The connected subgraphs formed by the links will display a modular structure. This would indicate that the networks formed by the participants' cross-linking of knowledge units display an emergence of complex organizing structures, indicating that the participants' crosslinking acts as a self-organiztion mechanism. We also hypothesized that discussions will vary in the extent of these properties. This would indicate that the subsequent network topology is related to the specific characteristics of each community and the content domain.

The combination of H1 and H2 was designed to support the claim that the cross-links introduced by Q&A community participants form organizing structures within the discussion

network that may facilitate navigation and so create a unique form of information network. If the cross-links affect the network structure in a way that contributes to its defragmentation and introduces emergent internal hierarchies this could imply that the participants collectively engage in a collaborative bottom-up structuring of the virtual knowledge collaboration environment.

To further address the emergent nature of the network structures, we analyzed the topological attributes of the discussion graphs at weekly intervals and compared them to the distribution of indicators extracted from multiple iterations of simulated respective null models with randomized cross-links. This comparison allowed for the monitoring of the effects of the cross-links on the discussion network as it evolves and might provide insight into the self-organizing quality of the networked discussions. Consequently, we hypothesized that:

H3: Over time, with the progression of the discussion, the disparity between the topological parameters of the real discussion graphs and their respective null models with randomized cross-links, will tend to increase. This would indicate an ongoing process of organization of content into a collective artifact that derives from cross-linking activity.

## Material and methods

### The Stack Exchange community discussions

The study examined online multi-participant discussions from Stack Exchange [42]. Stack Exchange is a network of more than a hundred question-answering (Q&A) topic-specific personal interest communities, covering a wide variety of topics such as History, Physics, Earth Science, Parenting, Beer, and many more. The largest and most known Stack Exchange community, called Stack Overflow, is centered around programming. Overall, the platform engages millions of people in posting questions, while others answer them asynchronously. Communities vary in size, but many include thousands of content-contributing participants and many thousands more passive participants, and span over nearly a decade. Researchers have studied Stack Exchange sites, and especially Stack Overflow, for various purposes. Among those, data from the discussions were used to discover the evolution of topic trends in a developer community [43, 44], to predict answer quality [45] or user participation [46], to identify experts [47], to recommend solutions to programming errors [48], and to analyze social interactions inside the cooperative community [49, 50]. The collaborative tagging system applied within the communities and its implications have also been the focus of previous work [17, 26, 27].

The data from eight Q&A discussions were downloaded from the Stack Exchange Data Dump, which hosts the entire history of every Stack Exchange community on August 24, 2021 [51]. These datasets, stored in XML format, include questions, answers, and comments, and also include specific information about linked questions: questions that the participants noted as referring to each other, by embedding hyperlinks in their answers. The data was converted to CSV format using RStudio (2020). The collection and analysis method complied with the terms and conditions for the use of Stack Exchange Data Dump data under the cc-by-sa 4.0 license, with specific limitations listed on the Stack Exchange Data Dump site. The datasets can be found at the OSF registries at https://doi.org/10.17605/OSF.IO/9XVNM.

The communities were selected arbitrarily but with some guidelines: First, all eight of the communities revolve around a shared interest in a knowledge domain ("earth science", "history"), as opposed to some communities that revolve e around a shared practice (e.g. "graphic design", "beer brewing"). Second, the communities that were selected ranged between 5000 and 25000 questions and spanned over at least five full years. The data used here was downloaded in August 2021 and includes the entire content of each discussion from its onset

**Table 2. Community discussion stats.**

| Topic | Onset | Duration (days) | Number of participants | Number of knowledge units | No. of cross-links [cross-link: node ratio] | % of linked knowledge units |
|---|---|---|---|---|---|---|
| Politics | 04/12/2012 | 3098 | 3853 | 12776 | 3378 [0.26] | 33.5% |
| Philosophy | 05/04/2011 | 3708 | 6573 | 16003 | 4070 [0.25] | 28% |
| Psychology and Neuroscience | 06/06/2011 | 3645 | 3784 | 7277 | 1790 [0.25] | 28% |
| Earth Science | 15/04/2014 | 2615 | 2998 | 5650 | 1018 [0.18] | 24% |
| History | 05/05/2011 | 3678 | 5273 | 12829 | 1865 [0.14] | 21% |
| Health | 31/03/2015 | 2551 | 3810 | 7189 | 669 [0.09] | 13.5% |
| Biology | 14/12/2011 | 3657 | 11190 | 25986 | 3749 [0.14] | 19.5% |
| Economics | 18/11/2014 | 2385 | 5510 | 11710 | 1135 [0.1] | 14% |

(determined by the date of the earliest post) to June 7, 2021. The onsets of the different discussions that were analyzed vary, and range between 2011–2015. Table 2 presents the general properties of the discussions, including the community topic, the time of onset of the discussion (based on the date of the first questions posted), the discussion's duration (in days), the number of participants, the number of knowledge units comprising a question along with its set of answers and comments, the total number of cross-links and the node:link ratio, and the percent of knowledge units that are linked to at least one other knowledge unit.

## Methodology

**Constructing the network graphs.** Network graphs are composed of nodes, that represent objects or entities, and of links that connect between them. In this work, we constructed network graphs based on the data from eight Stack Exchange sites, each covering a specific domain. Each of these websites can be viewed as a discussion held within a specified domain of content, in which new questions (posts) are posted independently, while answers (and comments) are posted in response to questions. In constructing the network for the discussions that we analyzed, each question, along with its answers and comments was regarded as a single "knowledge unit" for analysis purposes [22], and these comprised the networks' nodes. Administrative posts were not included. The edges consisted of the cross-links that were embedded within any component of the knowledge units in the form of URLs that refer to other knowledge units *within the same site*. These links are documented in the site's data archive (see archives at https://archive.org/details/stackexchange). We refer to these as cross-links. Although the cross-links are created from one question to another, once they are added the website displays the questions as linked. A bi-directional navigation path between them is then formed, with each question directing to the other question's page. Once posted, navigation along graph posts through their links is bi-directional. Participants can move from a post to any other post connected to it by clicking, regardless of the original direction of the link. Accordingly, we decided to treat the network as undirected. On a more theoretical level, the association between concepts and ideas is a two-way street. This research views the discussion as a holistic product of interconnected knowledge and does not address the specific directionality of links between knowledge units.

Using the R igraph package [52], we constructed 8 undirected unweighted network graphs. A negligible number of edges were redundant, that is, the same two questions were connected more than once, probably by different participants. These cases were rare and removed from the graphs. Questions that were removed by the community were also excluded from the graphs. While participants define most cross-links as indicating questions to be "related", a minority of them are labeled "duplicates" (between 6–14% in the discussions we analyzed). The "duplicate" cross-links were preserved from a standpoint that views them as representing a specific kind of relationship between knowledge units, which is part of the network.

**The null-model graphs.** In network-based research, null models are a method for generating the patterns expected from the data in the absence of the process of interest, as a relevant point for comparison [53]. According to Farine, a good null model meets two demands: 1) the aspect of the network that is of most interest to us is randomized, while 2) the model should strive to maintain all other aspects of the data constant. In the case of the current work, the process of interest is the effect of cross-linking on network topology, as a collaborative mechanism for the emergent organization of knowledge. Therefore, the null models we used were created by replacing the "cross-links" that the participants formed, with random links. Notably, adding random links to a network affects topology, for instance in terms of connectedness and the forming of connected components [23]. Using such models as an anchor for comparison allowed us to extract the effects of the intentionally constructed links that are the result of participant activity. To better simulate the creation of the real networks, which were formed gradually, we extracted the size parameters of the real networks, i.e. the number of nodes and the number of links, for each day from the discussions' initiation. The randomly linked network graphs for each day were then constructed based on the graph from the prior day, by adding to the existing network nodes and edges equal in number to the daily addition of new nodes and cross-links, the difference being that the additional edges were assigned randomly. Importantly, cross-links in the discussions are not necessarily created simultaneously with nodes, as these consist of entire knowledge-units made up of a question and its set of answers, if it has any. Links are embedded within the answers, and these can be added to the question at any point. Meanwhile new questions are formed independently of answers, and not all question-answer knowledge units become linked at all. To mimic the process of cross-linking, pairs of nodes within the "daily" simulated graph were randomly selected to become linked. As in the case of the real cross-linked networks, new links could not be redundant: they could only be introduced between nodes that have not been previously linked. So, each simulated "day" produces a network that is based on that of the previous day. Accordingly, growth pattern of the randomly linked graphs approximately mimics that of the real ones, the difference being the random assignment of the new links as opposed to associative links added by participants. This process of creating same-sized graphs with randomly linked nodes was iterated 1000 times for each community's network, resulting in 1000 simulated networks per community. Each of these graphs was identical to the corresponding real graph in terms of size and number of links and differed only in the assignment of links within the network, as random connections replaced the real ones. Next, for each of the 1000 simulated networks, network metrics were calculated based on the topology of the network on the final simulated "day". This was repeated for each of the communities so that each real-network topological metric could be compared against an estimated sampling distribution that was created using bootstrap resampling based on the 1000 corresponding metrics extracted from the simulated null-models for each of the metrics. Next, 99% confidence intervals were generated for each sampling distribution, to indicate the interval within which each metric would be likely to fall within given a randomly linked network. The real metrics were then compared against this interval, as a metric falling beyond the confidence interval would indicate with a high probability (>99%) that

it would not likely have been produced within a randomly linked network of the same size, link distribution, and growth rate. In turn, this would enable us to reject the null hypothesis and to conclude that the cross-linked networks assume qualities of emergent organizing structures that are common in other real-world information networks. Using bootstrap resampling to create the estimated sampling distribution was a means of coping with the need to compare a single real-world exemplar (the cross-linked discussion network) with a relevant simulated distribution of randomly linked null-model, by creating a confidence interval based on iterated simulations [54].

**Constructing graph networks for the chronological analysis.** For each of the graphs, a series of "snapshots" were created, representing the discussion at the end of each full week since its initiation. Topological metrics were then calculated for each weekly "snapshot", similar to the previous section. The metrics included: the number of triangles, the maximal degree of a single knowledge unit, and the size of the largest connected component. Next, using the same method described above, 100 graphs with the cross-links between knowledge units replaced with random ones were generated for each week. The metrics were calculated for each of the randomized-links graphs, resulting in 100 weekly series for each metric.

## Results

### Network topology metrics

As explained above, for each of the discussions, 1000 different networks were generated, composed of an identical number of nodes and links, and with a similar daily development in terms of graph size and the number of links.

Indicators for the properties of the discussion network were calculated by comparing the real network metric to the distribution of the metrics produced from the randomized network graphs. Table 3 describes these metrics:

For each of the discussions, the following metrics were calculated using the R packages igraph [52] and CINNA [55]: the size of the largest connected component, number of triangles, global clustering coefficient, maximal degree, rate of nodes with a degree of four and over. These were calculated for each of the real discussions, and the corresponding 1000 network graphs with randomized cross-links. For each metric, an estimated sampling distribution

**Table 3. Network metrics.**

| Metric | Calculation |
| --- | --- |
| Global clustering coefficient | the rate of closed triplets out of all triplets of connected nodes in the network, or the rate of nodes that are linked to each other, given that they are both linked to a third node. The clustering coefficient is reflective of the graphs transitivity [32], meaning in this case the likelihood of a content-unit associated with two other content-units to become a triad linked by association. Note that if pairs of connected units are scarce, then this rate can be very high regardless of the total number of triads in the graph. |
| Number of triangles | The overall number of fully connected triplets, or closed triangles. This complements the global clustering metric as it captures the actual number of associated triads. |
| max. degree | The maximal number of cross-links for a single knowledge-unit node |
| Degree≥4 | The number of knowledge units (nodes) with at least 4 cross-links |
| size of the largest connected component | The size (number of nodes) of the largest weakly connected subgraph, i.e. the largest portion of the network in which all nodes can be reached from all other nodes |
| Nodes in large components | The percentage of knowledge units (nodes) that are part of weakly connected components consisting of at least 10 nodes |

Table 4. Network metrics and their corresponding confidence intervals (CIs).

| community | size of the largest component (% of nodes) | % of nodes in components ≥ 10 | Number of triangles | Clustering coefficient | max. degree | Degree ≥ 4 |
|---|---|---|---|---|---|---|
| **politics** | 322 (2.5%) [308, 359] | 11.35 [9.82,9.9] | 180 [0.12,0.33] | 0.11 [0.0001, 0.0003] | 18 [7.1, 7.4] | 252 [31, 31.9] |
| **philosophy** | 2261 (14.1%) [621, 651] | 14.96 [9.69, 9.76] | 200 [0.26, 0.34] | 0.06 [0.0002, 0.0003] | 35 [7.6, 7.8] | 406 [42,43] |
| **psychology and neuroscience** | 625 (8.6%) [355,373] | 11.13 [9.56, 9.66] | 173 [0.3,0.4] | 0.14 [0.0005, 0.0007] | 22 [7.1, 7.3] | 174 [21, 21.7] |
| **Earth science** | 41 (0.7%) [44,47] | 5.22 [3.89, 3.99] | 55 [0.07, 0.11] | 0.15 [0.0003, 0.0005] | 10 [5.7,5.8] | 56 [4.6, 4.9] |
| **history** | 79 (0.6%) [23, 24] | 3.48 [1.19, 1.24] | 49 [0.02, 0.05] | 0.096 [0.0001, 0.0002] | 9 [5.3, 5.4] | 75 [2.9, 3.2] |
| **health** | 53 (0.7%) [12.4,13.1] | 1.6 [0.15, 0.17] | 22 [0.01, 0.04] | 0.11 [0.0002, 0.0006] | 8 [4.3, 4.4] | 32 [0.4,0.5] |
| **biology** | 1392 (5.3%) [34.8, 36.6] | 7.18 [1.96, 2] | 529 [0.03, 0.07] | 0.08 [0.00004, 0.00009] | 85 [5.9,6] | 336 [9.5,10] |
| **economics** | 46 (0.4%) [16,17] | 222 [39.5, 42.9] | 33 [0.009, 0.03] | 0.08 [0.00007, 0.0002] | 20 [4.8, 4.9] | 47 [1,1.7] |

In brackets: 99% confidence interval for metric in randomized graphs

was created using bootstrap resampling, based on the sample of the 1000 randomized graphs. Next, the probability for the metric extracted from the real graph to be part of this sampling distribution was calculated. This was done using the R package *boot* [54, 56]. Table 4 presents the metrics of the real discussion networks, along with the confidence interval (CI) containing an estimated 99% of the distribution of the corresponding metrics from the randomized networks.

The findings presented in Table 4 point out the disparity between the real graphs' network metrics and the randomized null-model metrics, as most of the real-graph measures fall well outside their corresponding confidence intervals.

The difference in the topological metrics indicates that the networks shaped by the participants' cross-links differ in topological properties from randomly linked counterparts. These properties include:

1) A higher rate of clustering was indicated by an increased clustering coefficient as well as a very significant increase in the absolute number of formed triangles. In all the networks, the average number of triangles across the randomized graphs was near zero, meaning that in most of the randomized graphs, no fully connected triplets were formed. Meanwhile, in the real networks, the number of triangles ranged from 22 for the Health community, to 529 for the Biology community, and their number was proportionate to the network's size (Pearson's r = 0.86). The increased clustering suggests a micro-level organization of the discussion content. While the research did not include systematic content analysis, it is plausible that linking between two knowledge units that share a connection to a third unit points to the linking of units that are similar in content or revolve around a specific sub-topic. This would also include questions that were viewed as duplicated. Fig 2 presents two examples of such connections from the History community network.

2) Forming of **local focal points** (Fig 3), indicated by an increase in the maximum number of connections held by a single node in the real graphs compared to the random graphs, and by a higher rate of nodes with four connections or more. In all the networks, the most connected knowledge units in the real graphs had significantly more connections than their counterparts in the randomized graphs. This suggests a form of knowledge units that are more

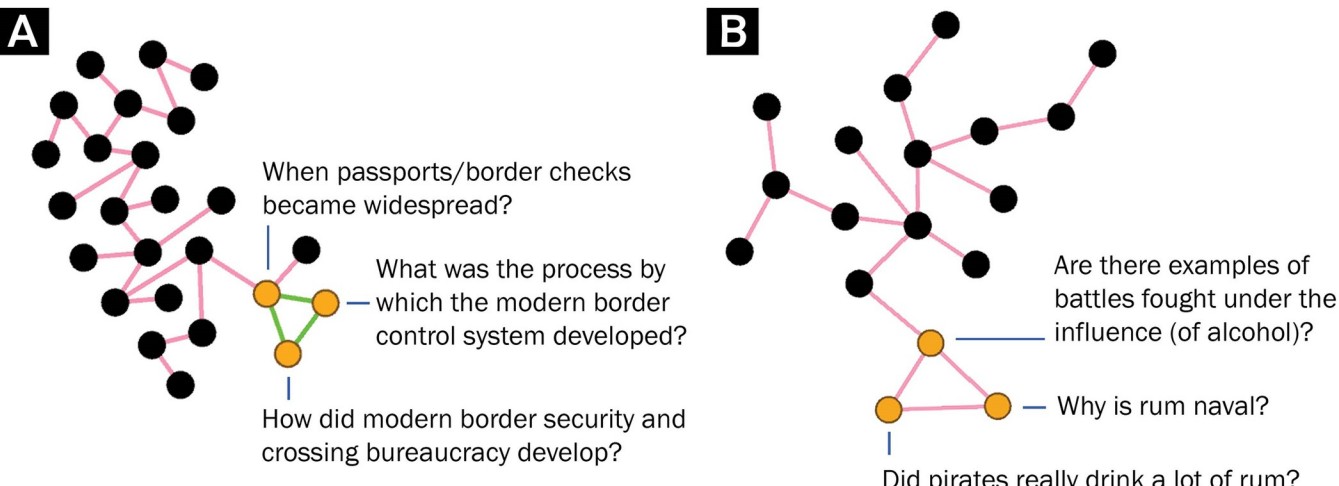

**Fig 2. Closed triplets.** Excerpts from the History community network, containing closed triplets, with the question titles. Green edges represent links indicating very similar knowledge units (i.e. "duplicated"). See the S1 Appendix for links to original posts (S1 Appendix).

connected and more central within the network than would have formed randomly, even in the case of a similar daily growth rate. These might be points of interest within the community, knowledge units that refer to more central concepts in the discussion, questions that were repeated multiple times and signaled as being duplicates, or any combination of these. The findings regarding the overall presence of more units with a degree of four and above imply that this was not limited to one central unit, but rather, to a phenomenon of multiple local focal points that formed across the network.

3) Defragmentation and convergence, indicated by an increase in the share of knowledge units that belong to connected components consisting of ten or more nodes, which was found in all eight discussions, and the forming of significantly larger connected components in comparison to the randomized networks, which were found in six of the eight discussions (Fig 4). These included the Economics network in which two large components of similar size were formed (one consisting of 46 nodes and the other of 43). Both were not likely to form within randomly linked networks of the same weekly size and density (the 99% confidence interval for the size of the largest connected component was: [16, 17]). The two remaining discussions included the Politics community and the Earth Science community. In the Politics discussion, the size of the largest connected component was within the confidence interval for the sizes of the largest components produced for the randomly linked graphs. This means that a connected component of similar size was likely to form for the same amount of daily knowledge units and cross-links introduced into the network, even if these were assigned randomly. In the Earth Science discussion network, the size of the largest component was significantly smaller than the size of the components found in the randomized networks. This indicates that the cross-links were not driving the network subgraphs to grow or converge into a larger graph as much as would occur within a network produced by adding random links to the network. Notably, the Earth Science community was the smallest of the eight communities observed, both in the size of the network of knowledge units and in the number of participants.

## Modularity

The modularity index was calculated for each network's largest connected component based on the Louvain algorithm [57], using the cluster_louvain function implemented in the igraph

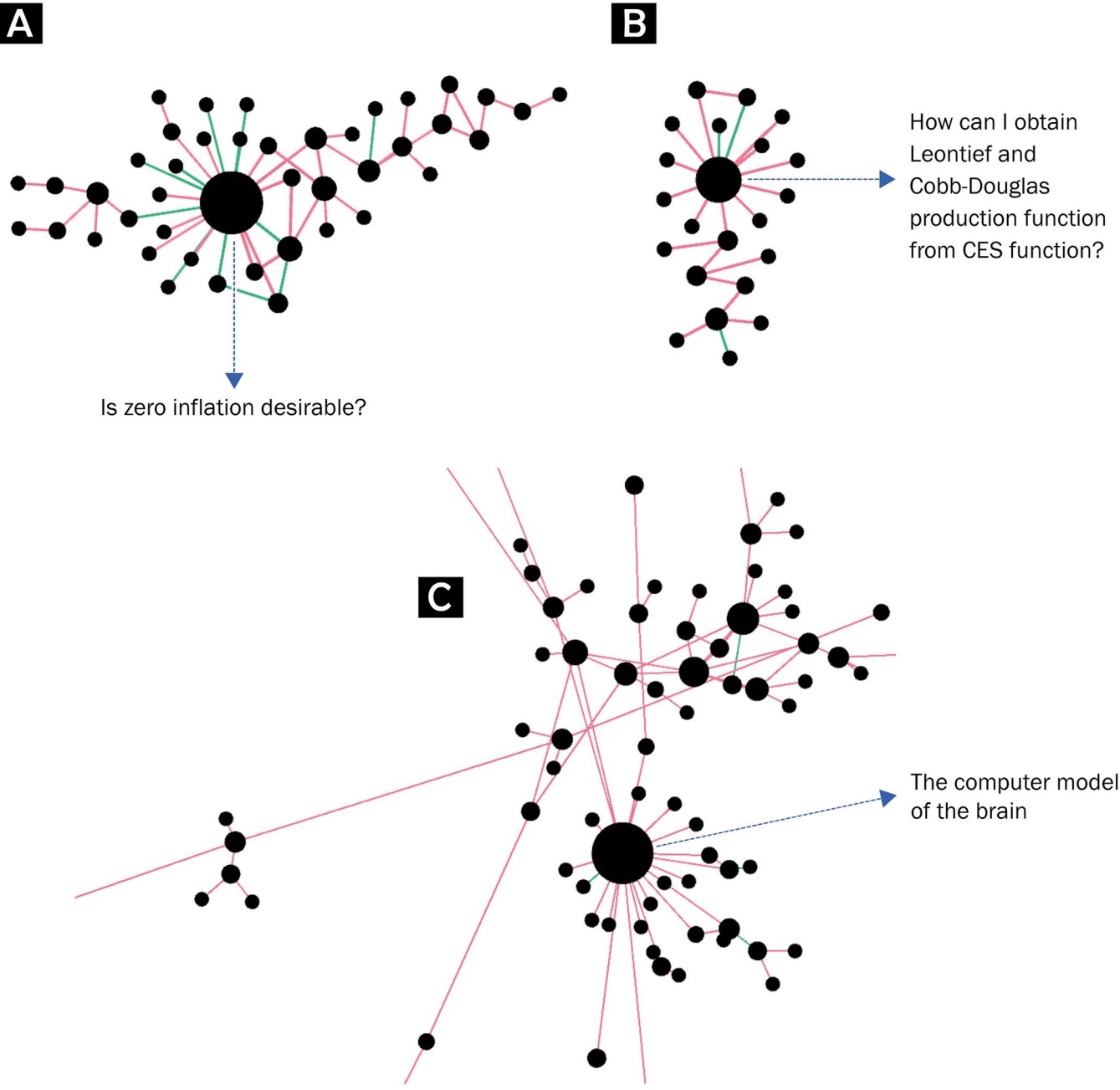

**Fig 3. Local focal points.** Fig 3(A)–3(C) present examples of such local focal points, from the Economics discussion and the Psychology and Neuroscience discussion. See the S1 Appendix for links to original posts (S1 Appendix).

R package. The analysis focused on large connected components as the modularity of the whole network is very high by definition, due to the networks' sparsity. The maximal-modularity index Q calculated by the Louvain algorithm ranges between -1 and 1, with values above zero indicating higher inner connectivity among nodes within modules than with nodes from other modules [58]. Table 5 features the modularity indices for the largest connected components from each network.

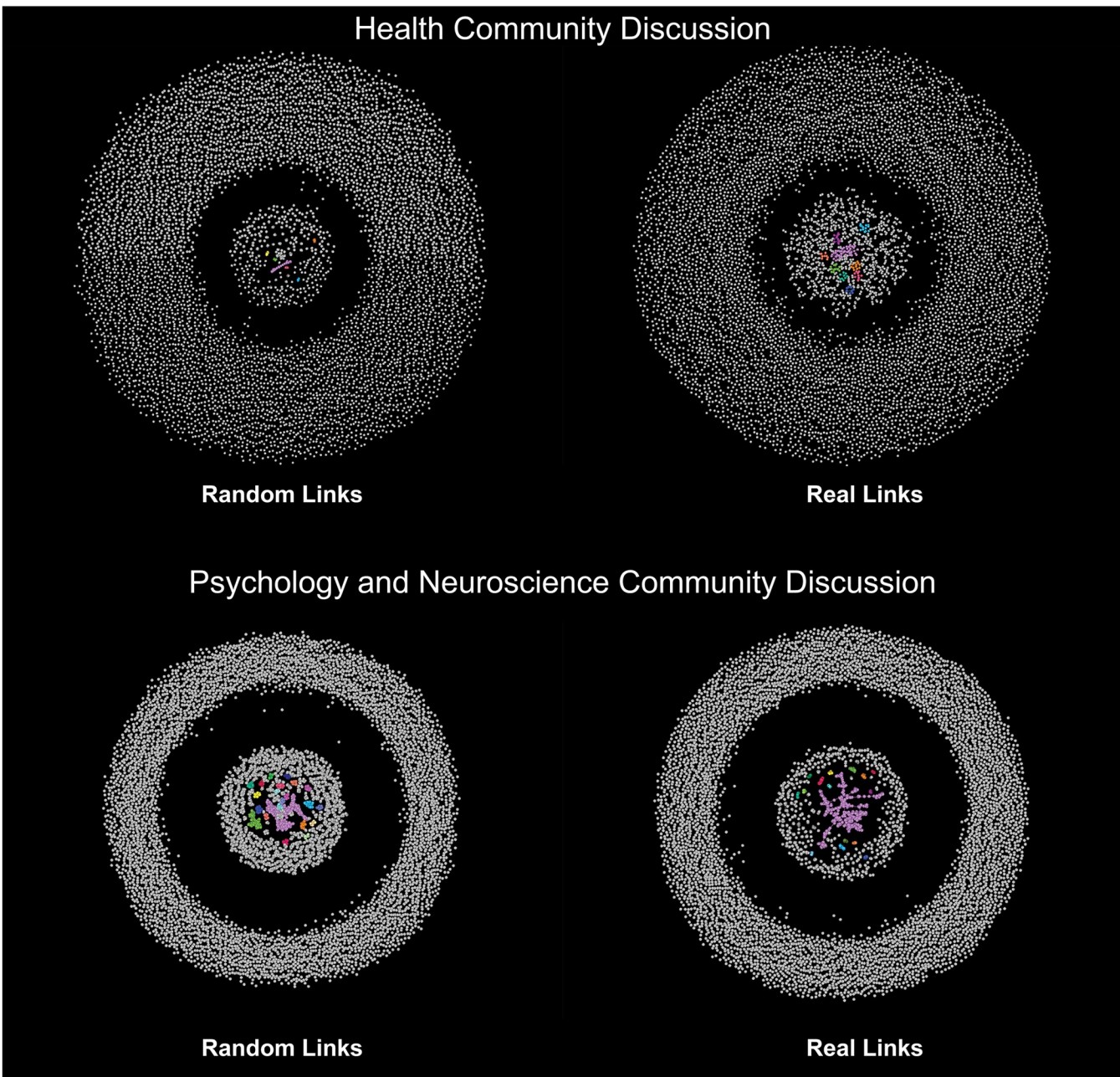

**Fig 4. Large connected components.** Components with 10 or more nodes in the real networks compared to networks with randomly produced links. In color: components of 10 nodes or more.

The findings presented in Table 5 indicate that the connected components maintain a highly modular inner structure.

Fig 5 displays the Philosophy network's largest connected component of 2261 nodes. The colors represent different modules.

## Chronological analysis findings

Findings regarding the difference in network metrics suggest that the real networks were different in topology compared to their counterparts with randomized cross-links. To assess

**Table 5. Modules and modularity of the largest connected components.**

|  | component size | number of modules | modularity |
|---|---|---|---|
| **politics** | 322 | 20 | 0.887 |
| **philosophy** | 2261 | 40 | 0.911 |
| **psychology and neuroscience** | 625 | 21 | 0.89 |
| **Earth science** | 41 | 6 | 0.636 |
| **history** | 79 | 8 | 0.751 |
| **health** | 53 | 8 | 0.7 |
| **biology** | 1392 | 38 | 0.876 |
| **economics** | 6 | 46 | 0.657 |

whether these differences grew, shrunk, or persisted over time and with the development of the discussion network, we used the weekly metrics that were described in the Materials and Methods section. These included calculations of the randomly linked networks' largest connected component, the maximal degree of a single node, and the number of closed triangles, at "weekly" intervals. Note that these were simulations of networks that were constructed based on the real weekly characteristics of the networks representing the Stack Exchange discussions. So, for each of the eight communities, for each of the three metrics, the data included 100 series of "weekly" observations. Each series was in the length, in weeks, of the duration of the discussion at the time the data was downloaded. This ranged from 322 for the shortest-lived community (Health) to 530 for the longest (Philosophy). All of the metrics increase with the growth of the networks. This is true for both randomly linked networks and real networks, with the exception that for some of the randomly linked networks, triangles were not formed at any

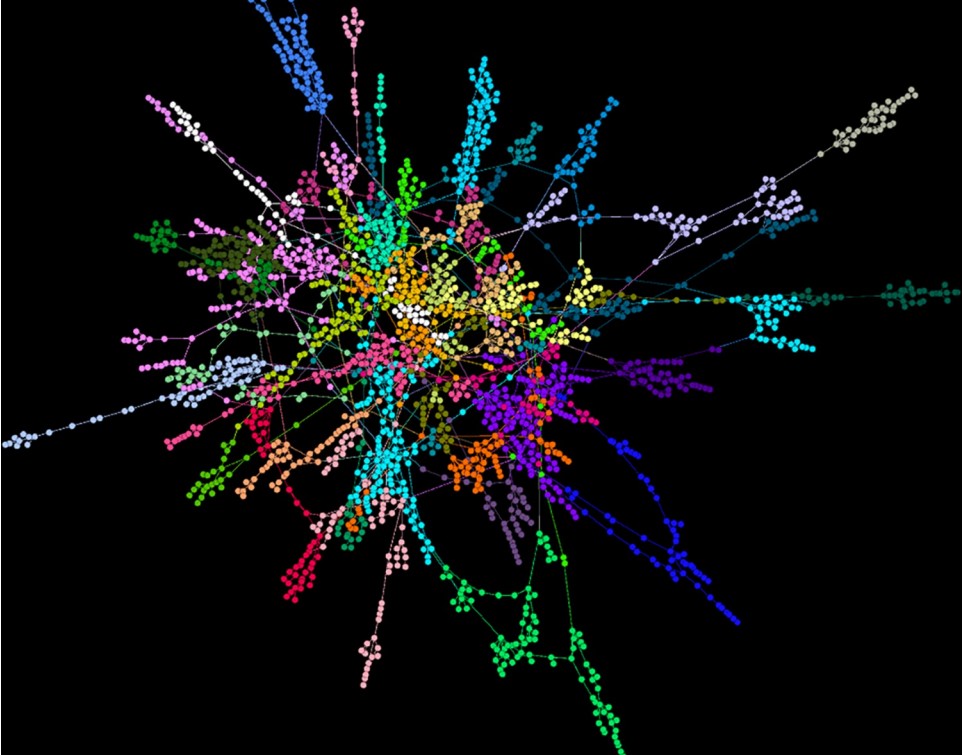

**Fig 5. The modular structure of the Philosophy network's largest connected component.**

**Table 6. Weekly growth regression coefficients for network metrics and the 99% confidence intervals for the regression coefficients of the randomly linked networks.**

| Community | No. of weeks | size of the largest component | max. degree | The number of triangles |
|---|---|---|---|---|
| | | regression coefficient [CI] | regression coefficient [CI] | regression coefficient [CI] |
| **politics** | 443 | 0.48 [0.47, 0.6] | 0.03 [0.008, 0.009] | 0.37 [0.0002, 0.0008] |
| **philosophy** | 529 | 4.41 [0.89, 1.12] | 0.067 [0.0074, 0.0086] | 0.38 [0.0002, 0.0007] |
| **psychology and neuroscience** | 520 | 1.22 [0.68, 0.79] | 0.037 [0.009, [0.01 | 0.31 [0.0002, 0.0007] |
| **Earth science** | 373 | 0.094 [0.098, 0.13] | 0.01 [0.006, 0.008] | 0.14 [0.0001, 0.0006] |
| **history** | 525 | 0.14 [0.035, 0.043] | 0.013 [0.006, 0.008] | 0.099 [0, 0.0002] |
| **health** | 323 | 0.123 [0.015, 0.019] | 0.016 [0.003, 0.004] | 0.056 [0, 0.0002] |
| **biology** | 523 | 3.34 [0.065, 0.079] | 0.2 [0.09, 0.01] | 1.061 [0.00002, 0.0003] |
| **economics** | 340 | 0.11 [0.026, 0.032] | 0.045 [0.004, 0.005] | 0.08 [0, 0.00007] |

* $p < 0.05$

**$p < 0.01$

point. However, in all the networks, triangles were formed in at least some of the randomly linked networks, so overall this too had a slightly positive trend. The point of the chronological analysis was to explore whether the growth of the metrics for the real networks differed significantly from that of the randomly linked networks. To test this, we analyzed the data in the following manner: based on the 100 weekly observations for each metric from the randomly linked networks, we created a generalized linear model with the bias-corrected and accelerated (BCa) interval [59]. This method generates a bootstrapped confidence interval for the model coefficient, by randomly sampling and resampling "individual subjects", calculating the individual regression coefficient, and using these to construct an estimated sampling distribution out of which a confidence interval for the model's coefficient is extracted. This non-parametric analysis is appropriate for longitudinal (hence dependent) data, as in our case. Following, the regression coefficients were calculated for the metrics drawn from the real networks. Table 6 displays the linear regression coefficients for each of these metrics' chronological week-by-week growth, and in brackets, the corresponding BCa 99% confidence interval for the coefficients of the randomly linked network metrics (CI). Higher coefficients indicate a steeper trendline or faster growth.

The regression coefficients for the changes over time in the metrics of the real networks were then compared to the simulated distribution of the corresponding coefficients for metrics from the randomly linked networks. This comparison revealed that the growth of the number of triangles and the maximal degree of a single node was, with high probability, faster in the real networks than would have been in randomly linked networks with a similar growth rate in terms of the number of nodes and links. This was found for every one of the eight communities in the study, indicating that the real networks tended to cluster more and produce specific nodes with more connections over time in comparison to randomly linked networks growing at the same pace in terms of the number of nodes and links.

As for the size of the largest connected component, this was found for six of the eight communities: the coefficients for the Philosophy, Psychology, and Neuroscience, History, Health, Biology, and Economics communities fell well above their corresponding CIs, indicating a faster growth of the largest component compared to those of the randomly linked networks. However, this was not the case for the Earth Science and the Politics communities, which also had a smaller largest component at the time of the data collection compared to their randomly linked counterparts. The coefficient for the Politics community's largest component linear growth rate fell within the CI for the randomly linked networks, while the Earth Science's

coefficient was slightly below the CI, indicating a slower growth of the largest component than would be expected from a randomly linked network of similar size and link density. A closer look at the growth of the largest components over time in the different communities reveals different patterns among communities: while in some communities the growth appears gradual and nearly linear, as a dominant large component grows by receiving new connected knowledge units, in other communities it grows in abrupt steps when two or more large components become connected at once within a linking peak. The Politics network belongs to the second category. An even closer look reveals that these peaks are correlated with election dates in English-speaking countries (especially the US but also the UK and Canada). Another such community is the Health community, from which a giant component emerged only in March 2020, with the rise of the COVID-19 pandemic. In addition, the weekly size of the largest component was determined by extracting the largest component from the network snapshot at the end of each week. While it is likely that for the most part, the same components gained volume and remained the largest ones throughout the progression of the discussion, it is possible that for some of the networks, two or more components were growing at similar rates, and hence the identity of the weekly largest component alternated between weeks, at least in the earlier weeks. Subsequently, it appears that the nature of the growth of large connected components and their relations to content and external occurrences needs to be approached specifically and receive more extensive research. Fig 6 displays the weekly growth of the real networks' largest connected components vs. the distribution of the largest connected components of the simulated randomly linked networks.

## Discussion

In this work, we used network structural qualities to examine the role of user-generated cross-links as a self-organizing mechanism in large online Q&A discussions. Our cumulative findings indicate that cross-links affect the structure of the discussion network, at the micro, mezzo, and macro levels and do so to a greater extent than would be predicted by a null model with random links replacing the genuine ones. The discussion networks tended to demonstrate a more complex topology compared to their randomized counterparts. This was indicated by increased clustering, the forming of larger connected components, and the emergence of more highly connected nodes in the networks that were generated based on the real discussion data, in comparison to networks that were constructed by replacing the cross-links with an equal amount of random links in a process that approximately mimics the formation of the discussion network in terms of growth. We further demonstrated that these effects are generally either maintained or enhanced over time and with the progression of the discussion and that their magnitude varies across communities and levels of the development process of communities. The connected subgraphs themselves were found to be of a highly modular structure. By comparing the cross-linked networks with corresponding randomly-linked networks, we were able to more effectively single out the impact of these purposeful links, compared to the general effects of the forming of an evolving network in terms of topology. Examining eight different communities operating on the same platform provided a further view of how different linking behaviors affect the topologies of the discussions, even within similar conditions in terms of the platform's affordances. The chronological analysis of the development of some of the network indicators over time provided another level for observation, and this analysis indicated that with the progression of the Q&A discussions, the difference between the real-graph networks and the randomized model networks tended to grow on all aspects measured. It also enabled observations on the relations between major events within the content domains and the topology of the discussions, which suggests that notable changes

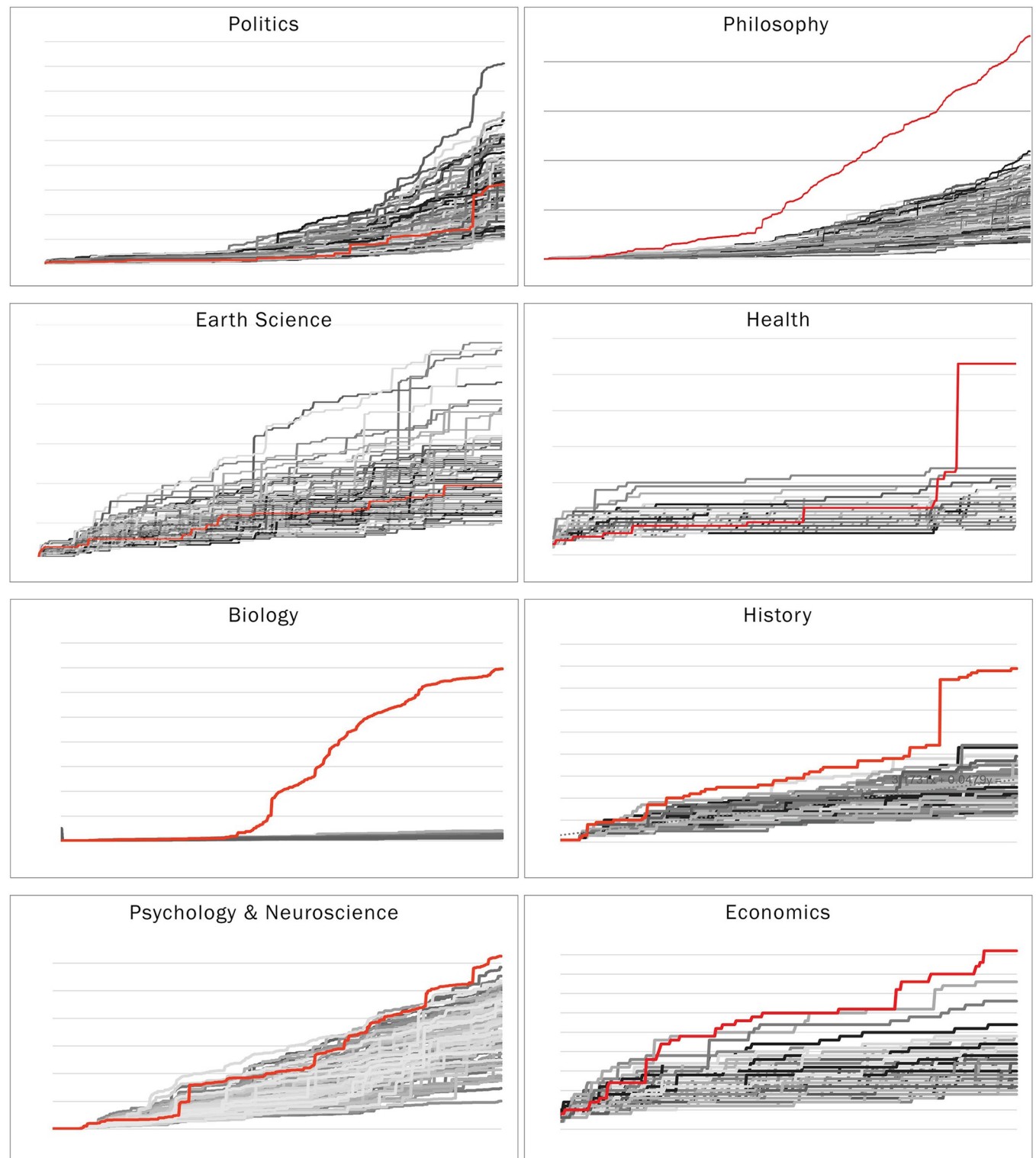

**Fig 6. Weekly growth charts of the networks' largest connected components.** The weekly growth of the largest connected components for each network. In red is the real network, and in shades of gray are the simulated random-link networks.

within the content domains (e.g. the eruption of the COVID-19 pandemic or major political events) are reflected in changes to discussion network topologies. The research adds to a growing literature that looks at Q&A discussions from a networked perspective, with a focus on the *artifact* layer of the network: the posts and the connections between them. The networks that were constructed were not social networks, in the sense that the network nodes consist of knowledge units and not of people. However, they were generated through the social activity of sharing knowledge, thoughts, and ideas regarding the connections between knowledge units. This emphasizes the important role that the participants' sharing of their own associative network by actively connecting knowledge units can play in shaping and structuring the collective knowledge product, in a manner that facilitates self-organization. It also highlights some of the possibilities brought on by addressing more aspects of the collaboration, that extend beyond contributing content on the one hand and interacting socially on the other.

## Limitations and directions for future research

Some limitations include the arbitrary selection of eight communities to analyze, which could be broadened to include a greater share of the 170+ communities operating within the Stack Exchange network, as well as other relevant platforms. Applying similar analyses to more communities could help to either generalize or identify constraints to the findings presented here and also to further explore the differences between communities. Another limitation is the focus on a single mechanism for self-organization, which is part of a more complex networked organizing system applied by the community that includes tagging as well as a text-based similarity algorithm that is applied by the platform. This was intentional, to highlight cross-linking as a mechanism that receives less attention. Still, future research could address the multiplex network that can be constructed by combining the different kinds of links, to formulate a more complete view of the knowledge network that emerges from the community's collaboration. This relates to another limitation which is the focus on structural and topological qualities detached from the ontological and semantic qualities of the discussion. Further research should explore the relationship between the two to deepen the understanding of the interplay between them. This was done to a certain extent by Ye et al. regarding content from the Stack Overflow website, but the analysis was restricted to the content network extending from a single exemplary 'seed' unit [22]. Future work could apply topic modeling methods for semantic analyses of entire discussion networks, and then compare these with the network's topology that is formed based on the cross-links. The cross-linked network itself could be elaborated to include the subnetworks composing the knowledge units, which include the question, answers, and comments, as a multi-layer network with two different types of edges–sequential links (question-answer-comment) and cross-links. Analyses of such networks would have to take into account the different types of nodes, i.e. questions, answers, and comments, and the different types of edges, i.e. cross-links vs. hierarchical sequential links.

As the study focused on the network of connected knowledge units, it is important to acknowledge that the majority of the knowledge units, in all of the discussions, did not become connected at all. The ratio of knowledge units that were linked to others ranged between 14% and 33.5% in the discussions that were examined. Many others were linked to only one other unit. This means that looking at the network formed by cross-links leaves out a large part of the content that was added by the community. Arguably, this could be viewed as a screening process that considers the knowledge units' linkage as some indication of their value to the conversation as a whole. Further work could try to address this point by examining the factors affecting a unit's likelihood of becoming connected at all, and specifically of becoming

connected to a larger component or even becoming a local focal point with multiple connected units. As explained in the Theory and Conceptual Framework section, the link-to-node ratio of the networks has implications for the emergence of organizing structures. More connected networks would potentially lead to more complex structures. However, while the question of artificially incentivizing cross-linking within the Stack Exchange environment using badges has come up, it was dismissed. It seems that while the community appreciates the cross-links and they are considered substantial contributions, the main reason given for withholding from gamifying cross-linking was that it will create an influx of links, which will make their added value diminish [60]. This makes sense from a network perspective, as it might impact the organic nature of the networks' development. Still, potentially raising the participants' awareness of the value of cross-linking, not through artificially incentivizing but through modifying community norms and socialization might contribute to the forming of effectively organized content networks. Potentially, this point could be studied within these communities through collaborations with community moderators.

In the current design of the Stack Exchange discussions, the users have access to the benefits of the cross-links as "road signs", directing them to relevant posts within the discussion network. This is realized in two ways. The first is the hyperlink that the repliers themselves embed within their answers or comments. The other is more salient and also bi-directional: the platform displays all linked questions to the side of the knowledge unit of questions, comments, and answers currently visited, regardless of the directionality of the link. This enables users to navigate the network based on the pathways their peers created for them. The role of the cross-links in forming mezzo-level and macro-level network structures, however, is not expressed in the platform's user interface. Possibly, adding a feature that taps into this property and feeds the information about the current network structure back into the conversation through a visual associative network map of the discussion, could affect the ways users navigate it as well as the community's perception of their knowledge sharing as a collective product. It could also highlight the participants' role in organizing the shared content [16, 61]. These potential influences could be further explored. Aside from the potential effects of platform affordances that might have an impact on participant behavior, the role of external influences could also be further explored. The analysis of the growth of connected components over time pointed to the plausible possibility the discussion network's topology may also be affected by external advancements in the community's field of interest, with changes reflecting the emergence of new associations among subjects in real life.

Another intriguing direction to follow would be a further inquiry into the dissimilarities between different communities: what are some of the explanations for a higher or lower tendency to cross-link, as well as for changes in this tendency over time? This could be assisted by modeling and comparing the evolution of different discussions, and possibly creating a generalized model for this process. Exploring the motivations of participants for cross-linking could also help shed light on the process and its outcomes and understand its interplay with other aspects of knowledge-sharing behavior. For instance, there might be users who are more inclined to link than others. Previous work on smaller-scale discussions has suggested that similar to many online phenomena, a small percentage of participants have a significant influence on the shaping of the conversation using cross-linking [29].

The current work focused on links within a community. However, answers often contain links to external sources, either other communities within Stack Exchange or links to other websites. Creating an extended network that captures the connections among different communities as well as the external contexts of knowledge could be a fruitful direction.

## Conclusion

In the current work, we applied a network approach to the content network created within online multi-participant discussions. This approach was used to demonstrate some of the effects of individual participants' local actions within the network, which consist of posting and linking, on topological changes in the overall structure of the network. In eight different Q&A communities operating within the Stack Exchange platform, indicators for the emergence of complex structures were found. These point to heightened global cohesiveness and integration, as well as to both micro-level and mezzo-level organizations of the network. The work presented here highlights the role of cross-linking within large-scale online discussions, a largely overlooked form of interaction with knowledge.

The analysis applied to whole networks enabled a broader view of how local linking can have implications for the larger-scale organization of the discussion as a network. The findings suggest that by altering the network topology, the combination of posting and linking can act as a self-organizing mechanism that allows for the formation of complex, non-linear organizing structures. The findings endorse the perception of cross-linked posts within a virtual knowledge collaboration as a unique form of information networks [32], joining other recognized networks of information such as citation networks or recommender networks. It further strengthens the emerging perception that undermines the traditional distinction between the collective product itself and the activities that were undertaken to coordinate it [62]. So, while from the participant's point of view, cross-links can act as a tool to navigate through the discussion, by moving from one knowledge unit to another on a path laid by a knowledgeable peer, from the community's standpoint they may be a means for promoting the formation of a collective knowledge-network product. This highlights the role of sharing associative connections as yet another level of interaction with content in virtual knowledge collaborations for both path creation and collective knowledge organization. In an individual cognitive setting, the emergence of macro-level structures within networks of content via micro-level connections can be considered an indication of self-organization [24]. In a collaborative setting, these sorts of structures may operate as an apt basis for the emergence of novel knowledge and collaborative meaning-making [31]. From a broader perspective, the approach presented here is inspired by work on individual cognitive systems and applied to a collective product, formed through multiple interactions. This direction could be further developed to explore possible manifestations of other cognitive processes at the collective content level.

## Supporting information

**S1 Appendix. References for Stack Exchange questions mentioned in Figs 2 and 3.**
(DOCX)

## Acknowledgments

We would like to acknowledge the valuable contributions of Dr. Amit Rechavi to this research. Dr. Rechavi's expertise in network analysis and his input regarding the methodological aspects of the research were invaluable in the completion of this work. We would also like to thank Dr. Neta Gilat for her helpful statistical advice, and Dr. Carmel Kent for her valuable contribution to the conceptualizations that preceded the current work and inspired it.

## Author Contributions

**Conceptualization:** Noa Sher, Sheizaf Rafaeli.

**Data curation:** Noa Sher.

**Formal analysis:** Noa Sher.

**Methodology:** Noa Sher.

**Software:** Noa Sher.

**Supervision:** Sheizaf Rafaeli.

**Visualization:** Noa Sher.

**Writing – original draft:** Noa Sher.

**Writing – review & editing:** Noa Sher.

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
