## [Decision Letter · Decision Letter 0]

23 Jan 2023

PONE-D-22-31022Associative linking for collaborative thinking: self-organization of content in online Q&A communitiesPLOS ONE

Dear Dr. Sher,

Thank you for submitting your manuscript to PLOS ONE. After careful consideration, we feel that it has merit but does not fully meet PLOS ONE’s publication criteria as it currently stands. Therefore, we invite you to submit a revised version of the manuscript that addresses the points raised during the review process.

We look forward to receiving your revised manuscript.

Kind regards,

Qaisar Shaheen, Ph.D

Academic Editor

PLOS ONE

Journal Requirements:

2. In your Methods section, please include additional information about how your Stack Exchange dataset was collected and ensure that you have included a statement specifying whether the collection and analysis method complied with the terms and conditions for the source of the data.

Additional Editor Comments:

Revise the manuscript as per the reviewers Suggestions.

Reviewers' comments:

Reviewer's Responses to Questions

**Comments to the Author**

1. Is the manuscript technically sound, and do the data support the conclusions?

Reviewer #1: Yes

Reviewer #2: Yes

2. Has the statistical analysis been performed appropriately and rigorously? 

Reviewer #1: Yes

Reviewer #2: Yes

3. Have the authors made all data underlying the findings in their manuscript fully available?

Reviewer #1: Yes

Reviewer #2: Yes

4. Is the manuscript presented in an intelligible fashion and written in standard English?

Reviewer #1: Yes

Reviewer #2: Yes

5. Review Comments to the Author

Reviewer #1: The manuscript “Associative linking for collaborative thinking: self-organization of content in online Q&A communities” shows the results of an analysis conducted on a Q&A dataset. The work highlights the topological properties of the corresponding network, obtained by considering knowledge units as nodes and cross-links mentioning other knowledge units es edges. In particular, the obtained networks are compared with random networks used as null models, showing that the topology of real networks is not reflected by the randomized counterpart.

The measures that are used for comparison are the size of largest component, the existence of hubs, clustering, and modularity.

The results that the authors find are not surprising and totally in line with previous works. It is indeed largely known in network science that random networks do not present clustering or other topological structures that are usually found in real networks [1,2,3].

For this reason I find a bit banal hypotheses H1 and H2, there was no reason to expect a different result.

I want to highlight that I found nothing wrong in the article, it is just that it should not be presented as a new discovery but just an analysis on a new dataset that confirms results that have been previously found on several types of real networks.

I suggest to reorganize the text and reformulate the scientific question as looking for a confirmation of known results in the Q&A network too.

[1] Newman M. Networks. Oxford university press, 2018.

[2] Barabasi A. Network Science. Cambridge University Press, 2016.

[3] Latora V., Nicosia V., and Russo G. Complex networks: principles, methods and applications. Cambridge University Press, 2017.

Minor:

line 121: it seems that a sentence’s end is missing

Line 152: “organic bottom-up means” —> “organic bottom-up mean”

- Line 140: The sentence “the network consists of content units composed of question posts and their corresponding answers, connected by cross-links embedded by participants within their answers.” is not clear enough to explain how networks are built because the cross-links are explained only later, in the Methodology section.

Reviewer #2: The presentation of the article is commendable but used to show some deficiencies that must be rectified before the decision, which are

Concern # 1: The article's abstract must reflect the literature article's recommendation.

Concern # 2: The background is too lengthy. Try to compress it.

Concern # 3: Elaborate the concept with suitable diagrams as visual things elaborate the concept far better than the textual.

Concern # 4: Expand the scalability of the article by including the research gap with prior work

Concern # 5: Read and cite the following articles

1.“Work coordination and collaborative knowledge construction in a small group collaborative virtual task”

2.“Faraway, Not So Close: The Conditions That Hindered Knowledge Sharing and Open Innovation in an Online Business Social Network”

6. PLOS authors have the option to publish the peer review history of their article (what does this mean?). If published, this will include your full peer review and any attached files.

Reviewer #1: No

Reviewer #2: **Yes: **Dr. Rizwan ALi Shah

---

## [Author Response · Author response to Decision Letter 0]

29 Mar 2023

Editor:

Additional information and data statement were added (p. 11)

Reviewer 1: 

1. The results that the authors find are not surprising and totally in line with previous works. It is indeed largely known in network science that random networks do not present clustering or other topological structures that are usually found in real networks [1,2,3].

For this reason I find a bit banal hypotheses H1 and H2, there was no reason to expect a different result.

I want to highlight that I found nothing wrong in the article, it is just that it should not be presented as a new discovery but just an analysis on a new dataset that confirms results that have been previously found on several types of real networks.

I suggest to reorganize the text and reformulate the scientific question as looking for a confirmation of known results in the Q&A network too.

response: Thank you for this opportunity to clarify an important point regarding the goal of the research. 

The goal of this work what was to use this known quality of real networks to make the point that the linked posts do form information networks, and that this is enabled by the users' individual linking activity.

The hypotheses addressed the user generated links' ability to introduce organization to the otherwise fragmented knowledge within the environment and to highlight the role of crosslinks within this process.

To further clarify this point and address the properties of the cross-link networks within the context of real-world networks, we've revised the phrasing of the hypotheses (See p. 9-10) 

line 121: it seems that a sentence’s end is missing

response: 

Thank you, this was a leftover from a previous version and was removed. 

Line 152: “organic bottom-up means” —> “organic bottom-up mean”

response: Means as in means to an end. 

- Line 140: The sentence “the network consists

of content units composed of question posts and their corresponding answers,

connected by cross-links embedded by participants within their answers.” is not

clear enough to explain how networks are built because the cross-links are

explained only later, in the Methodology section

response: This is an excellent point, the sentence was rephrased to address and explain cross-links explicitly: 

In our case, the network consists of nodes comprising content units of question posts and edges based on cross-links, the hyperlinks embedded by participants within their answers that refer to other content units within the same discussion. 

Suggested citations:

1] Newman M. Networks. Oxford university press, 2018.

[2] Barabasi A. Network Science. Cambridge University Press, 2016.

[3] Latora V., Nicosia V., and Russo G. Complex networks: principles, methods and applications. Cambridge University Press, 2017.

response: Thank you for your highly relevant suggestions. We've integrated both Newman (2018) and Latora et al. (2016) into the manuscript. Barabasi (2016) was already cited within the paper. The additions are as follows:

Newman was cited in the introduction and in the conclusion by integrating the cross-linked discussions into Newman's classification of "networks of information" . An important insight into the role played by communities within networks from the work by Latora et al. (2017) was integrated into the manuscript 

Reviewer 2

The article's abstract must reflect the literature article's recommendation 

response: The abstract was revised, hopefully the revisions reflect your input. 

The background is too lengthy. Try to compress it.

response: The background has been shortened 

Elaborate the concept with suitable diagrams as visual things elaborate the concept far better than the textual.

response: We've added another figure to illustrate the concept of cross-links

Expand the scalability of the article by including the research gap with prior work

response: See revised version of the paragraph beginning on line 119 

Read and cite the following articles

1.“Work coordination and collaborative knowledge construction in a small group collaborative virtual task”

2.“Faraway, Not So Close: The Conditions That Hindered Knowledge Sharing and Open Innovation in an Online Business Social Network”

response: Thank you for the references, the papers have been integrated into the manuscript (p. 3 and p.5)

---

## [Decision Letter · Decision Letter 1]

14 Jun 2023

PONE-D-22-31022R1

Associative linking for collaborative thinking: self-organization of content in online Q&A communities via user-generated links

PLOS ONE

Dear Dr. Sher,

Thank you for submitting your manuscript to PLOS ONE. After careful consideration, we feel that it has merit but does not fully meet PLOS ONE’s publication criteria as it currently stands. Therefore, we invite you to submit a revised version of the manuscript that addresses the points raised during the review process.

I would like to emphasize the importance of addressing the points raised by reviewers 1 and 3 as minor revisions, as I genuinely believe that this will result in a manuscript of outstanding quality, which will be highly regarded and valued by the interdisciplinary community. Thoughtfully incorporating the reviewers' comments will help strengthen the potential of your article, making it even more impactful.

Reviewer 3, in particular, has identified critical issues that, if addressed to some extent, can significantly enhance the contribution of your work. I acknowledge the effort and dedication you have invested in this research and encourage you to consider these suggestions toward the manuscript's acceptance. I have full confidence that, by implementing the proposed improvements, the final outcome will be a high-quality manuscript of significant relevance.

I sincerely appreciate the ongoing efforts you will dedicate to enhancing the article, and I look forward to reviewing the revised version, which I am confident will meet the expectations of the scientific community.

We look forward to receiving your revised manuscript.

Kind regards,

Carlos Henrique Gomes Ferreira, Ph.D.

Academic Editor

PLOS ONE

Journal Requirements:

Reviewers' comments:

Reviewer's Responses to Questions

**Comments to the Author**

1. If the authors have adequately addressed your comments raised in a previous round of review and you feel that this manuscript is now acceptable for publication, you may indicate that here to bypass the “Comments to the Author” section, enter your conflict of interest statement in the “Confidential to Editor” section, and submit your "Accept" recommendation.

Reviewer #1: (No Response)

Reviewer #2: All comments have been addressed

Reviewer #3: (No Response)

2. Is the manuscript technically sound, and do the data support the conclusions?

Reviewer #1: Yes

Reviewer #2: Yes

Reviewer #3: Partly

3. Has the statistical analysis been performed appropriately and rigorously? 

Reviewer #1: Yes

Reviewer #2: Yes

Reviewer #3: No

4. Have the authors made all data underlying the findings in their manuscript fully available?

Reviewer #1: Yes

Reviewer #2: Yes

Reviewer #3: Yes

5. Is the manuscript presented in an intelligible fashion and written in standard English?

Reviewer #1: Yes

Reviewer #2: Yes

Reviewer #3: Yes

6. Review Comments to the Author

Reviewer #1: In my previous review I highlighted that the results reported in this article do not appear as new nor surprising, they are actually quite expected. I hence suggested to make this more evident from the paper. The authors agreed with that, but the only change that the authors made with respect to my concern was to add a couple of sentences to hypotheses H1 and H2, while I would have expected a more substantial change in how the story is told. I suggest to modify the abstract and the introduction too in this direction. Also, it would be useful if the changes in the text were highlighted so as to easily identify them.

I also want to clarify that the references that I mentioned in my review were only meant to sustain my thesis that the results are not new, not a suggestion to include them in the article.

Reviewer #2: (No Response)

Reviewer #3: The very idea of the manuscript can be addressed, in my opinion, as follows: How cross-links (hyperlinks embedded by participants within their answers that refer to other content units within the same discussion) can shape Q&A network topology? It is clearly trying to comprehend Q&A's real-world networks, by means of the Complex Network science: Since these features, once identified, can be study or classified by using tools of this theory. Following the text, one can reckons that cross-links can act not as shortcuts, but bridges among different Q&A's contents. These cross-links are created by a participant by associating two unrelated contents. Then, the null synthetic network used to compare results is a random network, which in turn, has a probability to attached link among nodes. So that, authors are associating cross-links to links place by chance on the synthetic network counterpart.

Points:

1) In the point of view of network theory it is a "meta-population" (in the network theory jargon), since inside each sub-network exists a number of nodes associated with questions, answers and comments, and links among them. It was not take into account. With effect, it was handled simply as nodes connected by chance with other nodes, meaning random links among different contents. I do not think it is surprisingly or highlighted findings. The mapping of a real-network into a random one, in this context, do not correspond to the usual modeling coming from network science.

2) Why do not consider a cross-link, in the sense connecting different sub-networks, in the context of a "Network of networks"? It will sounds to be a more natural mapping, since the connectivity distribution of each sub-network seems to have relevant/substantial role in the receiving attention by participants to "create" a cross-link with another relevant sub-network. Thus, each layer, corresponding to a network with specific content, whereas a cross-link connects different layers.

3) I do not believe if the concept of a "hub" can be used on a connectivity distribution where the major number of nodes has a single connection, whereas the so-called "hubs" has four links... In Methodology section are described eight web sites corresponding to eight domain (network) of contents. Nodes are related to questions, besides answers and comments posted in response to questions. I believe that a more interesting question should be "what is the mathematical description for the connectivity distribution of cross-links of a Q&A networked system?", for example.

For example, let take this sentence:

"In all the networks, the most connected content units in the real graphs had significantly more connections than their counterparts in the randomized graphs."

I really do not understand the novel information that a magazine's reader can take from this... "The random modeling do not extract the fundamental mechanism of a Q&A real-network. So, its not random, because participants do not choose by chance what they will response, question or comment."

Another weak point:

"These included the Economics network in which two large components of similar size were formed (one consisting of 46 nodes and the other of 43)"

I was expecting some statistical treatment to this lower number of elements. I my opinion, with no data treatment a comparison with an statistical ensemble, provided by 1000 elements (null model), is doubtful since no strong fluctuations can permeate the network realizations. Remember that for random networks the average degree <k> is approximately k, but in the real-network counterpart not necessarely.

With all this considered, I do not recommend the current version of the manuscript for publication. I do not believe the work is able to attract readers attention.</k>

7. PLOS authors have the option to publish the peer review history of their article (what does this mean?). If published, this will include your full peer review and any attached files.

Reviewer #1: No

Reviewer #2: **Yes: **Dr. Rizwan Ali Shah

Reviewer #3: No

---

## [Author Response · Author response to Decision Letter 1]

28 Jul 2023

Response to reviewers

We would first like to thank you all for taking the time and making the effort to review this paper. Your feedback has been instrumental. We realize that some major points within the manuscript were not clear enough and needed further clarification. Mainly, it is important to emphasize that this research aimed to enrich our understanding of the collective creation of knowledge, by addressing the largely overlooked perspective of the shared product of virtual knowledge collaborations as a wholistic unit. While this was assisted by network science tools, these were not the main thrust of the paper in terms of novelty. We set out to point to the emergent networked nature of collectively created organizing structures, based on shared linking, and not to create a mathematical model for this phenomenon. While this could be a fascinating direction for future work, it is beyond the scope of the current one. We believe that the current literature has not given enough attention to the concept of shared linking as a means of creating a collective networked product, and its significance as a networked phenomenon has to first be explored, introduced, and acknowledged, before further investigation. Accordingly, the paper's aims and scope were further clarified. See the revised abstract, introduction, and discussion sections. 

See more detailed responses to points raised within the reviews in the following table (all page numbers apply to the tracked-changes version of the manuscript):

Suggestion /Response

Reviewer 1: 

In my previous review, I highlighted that the results reported in this article do not appear as new nor surprising, they are actually quite expected. I hence suggested to make this more evident from the paper. The authors agreed with that, but the only change that the authors made with respect to my concern was to add a couple of sentences to hypotheses H1 and H2, while I would have expected a more substantial change in how the story is told. I suggest to modify the abstract and the introduction too in this direction. Also, it would be useful if the changes in the text were highlighted so as to easily identify them. 

Response: As you have pointed out, the findings regarding the topology of the cross-linked networks are not surprising given the assumption that these qualify as real-world networks. However, the main idea of this research is not to establish these well-known phenomena in the study of real-world networks but rather to harness them to demonstrate that these unique kinds of networks, which have not received much attention in the existing literature of virtual knowledge collaborations, can be regarded as a collective product in the form of a network. This had to first be established, and then further explored by examining the emergence of modularity, the emergence of organizing structures over time, and the qualitative insights regarding the relations between the network properties and concurrent real-world events. As this has not been clear enough in the previous versions of the manuscript, several changes were made, including a re-phrasing of the abstract and the research hypotheses in this direction. 

See the version with highlighted tracked changes, especially the abstract and the introduction sections, and page 12)

Reviewer 3 

1) In the point of view of network theory it is a "meta-population" (in the network theory jargon), since inside each sub-network exists a number of nodes associated with questions, answers and comments, and links among them. It was not take into account. With effect, it was handled simply as nodes connected by chance with other nodes, meaning random links among different contents. I do not think it is surprisingly or highlighted findings. The mapping of a real-network into a random one, in this context, do not correspond to the usual modeling coming from network science.

2) Why do not consider a cross-link, in the sense connecting different sub-networks, in the context of a "Network of networks"? It will sounds to be a more natural mapping, since the connectivity distribution of each sub-network seems to have relevant/substantial role in the receiving attention by participants to "create" a cross-link with another relevant sub-network. Thus, each layer, corresponding to a network with specific content, whereas a cross-link connects different layers. 

Response: As you correctly point out, the full network of content within the Q&A discussion indeed includes questions, answers, and comments, and so the cross-linked network could be viewed as a "network of networks" comprising the connected content units as an overarching network with the question-answers-comments components as sub-networks. However, these subnetworks are by design hierarchical tree-shaped networks, and so emergent topological structures, which were the focus of this work, cannot appear within them. This phenomenon occurs within the question layer, as questions, and not answers or comments, become linked. For this reason, the more simplified network of content units was fit for purpose. Future work could further dive into other properties such as the number of answers and comments as a metric for a content-unit's significance. This explanation has been added to the paper (see p.6, p.9 in the tracked changes version). 

As for the mapping of the network, the randomized simulations were not an attempt to find a mathematical equation describing the network. Their role was in creating an estimated sampling distribution, using Bootstrap, to form statistical confidence intervals for the randomly linked networks' metrics, which served as a basis for comparison with the real metrics. Each of these simulations included 1000 iterations, which were used for extracting the metrics and creating the estimated sampling distributions. Importantly, we would like to emphasize here that the current work was not aimed at creating a model of such a network but rather to establish the "networkedness" of the collective artifact, by demonstrating that it harbors common real-world network properties that point to self-organization, as opposed to a random formation. This point has been added to the paper, with a more detailed explanation of the statistical process which we agree was not explained properly in the previous version (see changes in abstract, see abstract, introduction and methodological sections).

3) I do not believe if the concept of a "hub" can be used on a connectivity distribution where the major number of nodes has a single connection, whereas the so-called "hubs" has four links... In Methodology section are described eight web sites corresponding to eight domain (network) of contents. Nodes are related to questions, besides answers and comments posted in response to questions. I believe that a more interesting question should be "what is the mathematical description for the connectivity distribution of cross-links of a Q&A networked system?", for example.

For example, let take this sentence:

"In all the networks, the most connected content units in the real graphs had significantly more connections than their counterparts in the randomized graphs."

I really do not understand the novel information that a magazine's reader can take from this... "The random modeling do not extract the fundamental mechanism of a Q&A real-network. So, its not random, because participants do not choose by chance what they will response, question or comment." 

Response: We acknowledge your comment on the use of the concept of "hubs". Its use was an adaptation, as the cross-linked networks are very sparse, which is an artifact of the domain, and so what would commonly be referred to as "hubs" would be extremely rare, if possible at all. We added a paragraph that explains this and changed the wording accordingly (see p.9). 

As for the mathematical equation, as mentioned, this was not the focus of the current work, and its significance is largely for literature on collective cognition and the collaborative creation of knowledge than for literature in network science. The main point was to demonstrate how individual activity that consists of sharing links results in changes to the overall structure of the network, which can be considered as organizing process. Future work could dive deeper to discover a mathematical description for this process, and perhaps use that to compare different discussions as there seems to be notable variance between communities which is worth exploring. 

Another weak point:

"These included the Economics network in which two large components of similar size were formed (one consisting of 46 nodes and the other of 43)"

I was expecting some statistical treatment to this lower number of elements. I my opinion, with no data treatment a comparison with an statistical ensemble, provided by 1000 elements (null model), is doubtful since no strong fluctuations can permeate the network realizations. Remember that for random networks the average degree is approximately k, but in the real-network counterpart not necessarily.

response: The description of the statistical procedure was revised and further clarified to address this point. The 1000 iterations of different randomly linked networks per each real network, which were similar in size and growth rate, network were used for extracting 1000 corresponding metrics for each of the networks, which act as a sample of randomly linked networks of similar size and growth rate. These 1000 "observations" for each of the metrics for each of the networks were used for creating an estimated sample distribution through bootstrap resampling which was used for creating a 99% confidence interval for each of the metrics. Falling beyond this interval demonstrates a high probability of not being the product of random linking, which strengthens the hypothesis that the cross-links introduce emergent order into the otherwise unorganized (and fragmented) discussions (see additions in p 12 and 17).

---

## [Decision Letter · Decision Letter 2]

16 Oct 2023

PONE-D-22-31022R2Associative linking for collaborative thinking: self-organization of content in online Q&A communities via user-generated linksPLOS ONE

Dear Dr. Sher,

Thank you for submitting your manuscript to PLOS ONE. After careful consideration, we feel that it has merit but does not fully meet PLOS ONE’s publication criteria as it currently stands. Therefore, we invite you to submit a revised version of the manuscript that addresses the points raised during the review process.

We look forward to receiving your revised manuscript.

Kind regards,

Carlos Henrique Gomes Ferreira, Ph.D.

Academic Editor

PLOS ONE

Reviewers' comments:

Reviewer's Responses to Questions

**Comments to the Author**

1. If the authors have adequately addressed your comments raised in a previous round of review and you feel that this manuscript is now acceptable for publication, you may indicate that here to bypass the “Comments to the Author” section, enter your conflict of interest statement in the “Confidential to Editor” section, and submit your "Accept" recommendation.

Reviewer #1: All comments have been addressed

Reviewer #3: All comments have been addressed

Reviewer #4: (No Response)

2. Is the manuscript technically sound, and do the data support the conclusions?

Reviewer #1: Yes

Reviewer #3: Partly

Reviewer #4: Partly

3. Has the statistical analysis been performed appropriately and rigorously? 

Reviewer #1: I Don't Know

Reviewer #3: No

Reviewer #4: Yes

4. Have the authors made all data underlying the findings in their manuscript fully available?

Reviewer #1: Yes

Reviewer #3: Yes

Reviewer #4: Yes

5. Is the manuscript presented in an intelligible fashion and written in standard English?

Reviewer #1: Yes

Reviewer #3: Yes

Reviewer #4: Yes

6. Review Comments to the Author

Reviewer #1: I would have appreciated if the authors had highlighted in a different color the changes that they made to the text. Anyway they have modified and improved the text so I suggest acceptance.

Reviewer #3: I am not convinced by the answers provided by the authors... These answers are essentially devoted to including some explanations in the text. For example:

"We acknowledge your comment on the use of the concept of "hubs". Its use

was an adaptation, as the cross-linked networks are very sparse, which is an artifact of

the domain, and so what would commonly be referred to as "hubs" would be extremely

rare, if possible at all. We added a paragraph that explains this and changed the

wording accordingly (see p.9)."

So, essentially the authors' responses was an "adaptation" of what reviewers have pointed out. It causes a feeling like "we are speaking to the reader with the complex network language, but not quite".

Another interesting point can be highlited with the following example:

"As for the mathematical equation, as mentioned, this was not the focus of the current

work, and its significance is largely for literature on collective cognition and the

collaborative creation of knowledge than for literature in network science. The main

point was to demonstrate how individual activity that consists of sharing links results in

changes to the overall structure of the network, which can be considered as organizing

process. Future work could dive deeper to discover a mathematical description for this

process, and perhaps use that to compare different discussions as there seems to be

notable variance between communities which is worth exploring."

I am convinced that this work should be sent to another journal with a profile more focused on "studies of cases", since the text does not seem to be concerned with reproducibility or revealing fundamental mechanisms of the studied phenomenon. Plus, my opinion is the opposite of that of the authors: not including an universal language, such as mathematics, means attracting the attention of a restricted group of readers.

In view of these considerations, I do not recommend the manuscript because do not believe that it is able to attract the attention of the magazine's readers.

Reviewer #4: The submission "Associative linking for collaborative thinking: self-organization of content in online Q&A communities via user-generated links" presents an analysis of the clustering, modularity, connectivity, and percolation of a suite of eight networks based on the StackExchange (SE) forums. These networks were defined on the set of threads--referred to here as "content units"--and edges were drawn between them based on whether or not a cross-link exists anywhere in the question, answer, or comments of the thread between thread x and thread y. The measurements on these networks are compared to what I would describe as a percolated hypercanonical Erdos-Renyi (ER) model in which the probability of an edge (or the number of links) is pulled from a distribution which matches the number of links created from each day in the data. This is not done in any sort of Monte-Carlo way, it is meant to match the exact percolation rate of each day seen in the data. The results show that the Stack Exchange networks exhibit clustering, max degree, nodes of degree >= 4, and largest component size to be outside of the 99% CI for 1000 runs of this percolated hypercanonical ER null model. They find also that the largest connected components exhibit high modularity and growth in the largest degree and number of triangles over time exceeds the rates seen in the null model.

It is my opinion that the strongest point of this article is that they have performed network analysis upon a knowledge-creating system which I have not seen an analysis of yet, and they do well to perform the relevant literature review.

However, there are a number of shortcomings in this paper which must be overcome for it to be ready to publish.

1a. The focus of the introduction and the motivation of this article seems to be to prove that these cross-link SE networks are "real networks." The authors state: "...this would enable us to reject the null hypothesis and to conclude that the cross-linked networks assume qualities that resemble real-world networks." [385-386] which is an odd hypothesis to have because there is no objective criterion for "real networks." A "network" is an ontological object and a "real network" would just be a real system which is examined through a graphical model. It seems that a lot of the conclusions about "emergent organization" rely on some proof that this cross-linked network is a "real network." It would be more salient to frame this as simply examining the features of this network (which is inherently a "real network" because it exists in the world and is being modeled graphically) and suggesting what that says about the system.

2. This work has little to no basis in social-scientific theory about knowledge-contribution behavior, so I am uncertain as to what we have learned from this study besides that the cross-link network likely has more internal structure than a random graph. The question remains: what are we to make of these result? In response to another reviewer who asked a similar question, I see that the response was that the goal is to bring attention “the concept of shared linking as a means of creating a collective networked product” but none of the papers cited (as far as I have seen) lend any meaning to this term “collective networked product” or its significance, let alone how these network properties map to this concept. The paper cites Barabasi’s 2003 paper to suggest that clustering implies organizing (also it may be worth considering Krioukov’s 2016 PRL paper) but this is very vague. Even a step forward to citing Newman’s definition in his 2005 textbook that ‘clustering implies transitivity’ would lend at least some theoretical advancement–-i.e. Cross-linking behaviors exhibit some transitivity.

3. There are a number of issues with the null model. It is a bit of an unusual null model to use because it seems to be a percolated G(n,p)/G(n,m) model with the link parameter matching the exact number of each day. The result of this would, ultimately however, still be a G(n,p) model with p=sum(edges)/sum(nodes) at the end. As this scales independently of the number of nodes, this is a dense model (see Def. 1.3 in Random Graphs and Complex Networks by Remco van der Hofstad) and thus I’m not sure this is the appropriate null model to measure the temporal dynamics of the network.

Other smaller things:

a. Including triangles and clustering is redundant

b. I'm uncertain as to why the terms "content units" and "focal points" are introduced and what purpose they serve. It seems that "content unit" refers to a thread in the forum, and “focal unit” is referring to a node of degree >= 4 but there is no justification as to why that was chosen, nor any theory behind it to suggest the significance of such in knowledge-sharing systems.

c. Sometimes the word “sparse” gets used to seemingly describe how disconnected the network is. The ratio presented of links to nodes is actually dense enough to have one connected component and is dense in comparison to many other networks of real systems. In general, however, we measure sparsity and density within-component (so infinite distances get dropped).

Broadly, my recommendation would be to zoom in and reframe the work so that the genuinely good results obtained can be understood through the lens of theory behind knowledge-sharing, instead of trying to develop novel theory about the relationships between network topology and epistemic knowledge-units alongside obtaining results.

7. PLOS authors have the option to publish the peer review history of their article (what does this mean?). If published, this will include your full peer review and any attached files.

Reviewer #1: No

Reviewer #3: No

Reviewer #4: **Yes: **Sagar Kumar

---

## [Author Response · Author response to Decision Letter 2]

11 Jan 2024

Rebuttal letter

We would like to thank all of the reviewers who have taken their time and made the effort to read our work and make suggestions. Your contributions have been instrumental.

As reviewer 1 had no further comments in this round, reviewer 2 has not added any further comments after the first round, and reviewer 3 has indicated that they disagree with the concept of the paper, this response addresses the suggestions made by reviewer 4. Please see below:

Reviewer #4: The submission "Associative linking for collaborative thinking: self-organization of content in online Q&A communities via user-generated links" presents an analysis of the clustering, modularity, connectivity, and percolation of a suite of eight networks based on the StackExchange (SE) forums. These networks were defined on the set of threads--referred to here as "content units"--and edges were drawn between them based on whether or not a cross-link exists anywhere in the question, answer, or comments of the thread between thread x and thread y. The measurements on these networks are compared to what I would describe as a percolated hypercanonical Erdos-Renyi (ER) model in which the probability of an edge (or the number of links) is pulled from a distribution which matches the number of links created from each day in the data. This is not done in any sort of Monte-Carlo way, it is meant to match the exact percolation rate of each day seen in the data. The results show that the Stack Exchange networks exhibit clustering, max degree, nodes of degree >= 4, and largest component size to be outside of the 99% CI for 1000 runs of this percolated hypercanonical ER null model. They find also that the largest connected components exhibit high modularity and growth in the largest degree and number of triangles over time exceeds the rates seen in the null model.

It is my opinion that the strongest point of this article is that they have performed network analysis upon a knowledge-creating system which I have not seen an analysis of yet, and they do well to perform the relevant literature review.

However, there are a number of shortcomings in this paper which must be overcome for it to be ready to publish.

1a. The focus of the introduction and the motivation of this article seems to be to prove that these cross-link SE networks are "real networks." The authors state: "...this would enable us to reject the null hypothesis and to conclude that the cross-linked networks assume qualities that resemble real-world networks." [385-386] which is an odd hypothesis to have because there is no objective criterion for "real networks." A "network" is an ontological object and a "real network" would just be a real system which is examined through a graphical model. It seems that a lot of the conclusions about "emergent organization" rely on some proof that this cross-linked network is a "real network." It would be more salient to frame this as simply examining the features of this network (which is inherently a "real network" because it exists in the world and is being modeled graphically) and suggesting what that says about the system. 

Authors' response:

Thank you for your input. Since as you mentioned, this type of information network has scarcely been addressed in the literature, we wanted to first establish that the collection of content-units (question and answer sets) connected by user-generated internal references constitutes as a network with a structure worth exploring. The wording was revised to better describe this. 

2. This work has little to no basis in social-scientific theory about knowledge-contribution behavior, so I am uncertain as to what we have learned from this study besides that the cross-linked network likely has more internal structure than a random graph. The question remains: what are we to make of these result? In response to another reviewer who asked a similar question, I see that the response was that the goal is to bring attention “the concept of shared linking as a means of creating a collective networked product” but none of the papers cited (as far as I have seen) lend any meaning to this term “collective networked product” or its significance, let alone how these network properties map to this concept. The paper cites Barabasi’s 2003 paper to suggest that clustering implies organizing (also it may be worth considering Krioukov’s 2016 PRL paper) but this is very vague. Even a step forward to citing Newman’s definition in his 2005 textbook that ‘clustering implies transitivity’ would lend at least some theoretical advancement–-i.e. Cross-linking behaviors exhibit some transitivity.

Authors' response:

The theoretical concept behind the paper is highly connected to knowledge-contribution behavior: the internal complex structures present a form of self-organization (see for instance Heylighen (in Bates & Maack. eds), 2015). This structuring is suggested as form of knowledge integration that exceeds the integration created within the text or by other means of structuring virtual knowledge collaborative discussions such as tagging. This point has now been made much more salient within the text, especially in the introduction and theoretical concept sections.

3. There are a number of issues with the null model. It is a bit of an unusual null model to use because it seems to be a percolated G(n,p)/G(n,m) model with the link parameter matching the exact number of each day. The result of this would, ultimately however, still be a G(n,p) model with p=sum(edges)/sum(nodes) at the end. As this scales independently of the number of nodes, this is a dense model (see Def. 1.3 in Random Graphs and Complex Networks by Remco van der Hofstad) and thus I’m not sure this is the appropriate null model to measure the temporal dynamics of the network.

We have given precise mathematical definitions for the notions of graph sequences being highly connected, small worlds, and scale free, extending earlier definitions in van der Hofstad (2010). Our definitions are based upon a summary of the relevant results proven for random graph models. We restrict ourselves to sparse random graphs, i.e., random graphs where the average degree remains bounded as the network size grows (recall Definition 1.3). In recent years, there has been an intensive and highly successful effort to describe asymptotic properties of graphs in the dense setting, where the average degree grows proportionally to the network size. This theory is described in terms of graph limits or graphons, which can be thought of as describing the limit of the rescaled adjacency matrix of the graph. The key ingredient in this theory is Szemer´edi’s regularity lemma (see Szemer´edi (1978)), which states roughly that one can partition a graph into several more or less homogeneous parts with homogeneous edge probabilities for the edges in between. See the book by Lov´asz (2012) for more details. We refrain from discussing the dense setting in more detail.

Authors' response:

The number of both edges and nodes in the simulated graphs are identical to those of the original graphs, and were constructed as follows: the real graph's daily addition of nodes and edges was extracted, and the randomly-linked simulated graphs were constructed by gradually adding the daily number of nodes and edges while the structure of the graph remains. That is, nodes were added and then edges were randomly assigned on top of the existing structure from the previous simulated "day" (without repetition, only nodes that have not been linked could become linked). So, at the end of each simulated "day", there is a graph identical in a number of both nodes and links to the snapshot of the real graph on that day. On the "final day", the number of nodes and edges is identical to that of the real graph, and the average degree is of course identical. This was done to count for the greater chance of earlier nodes becoming connected, as the simulated graphs mimic the growth patterns of the real ones. This was iterated 1000 times so there eventually are 1000 simulated graphs for each community. So, the difference between the real graphs and their simulated counterparts is the assignments of edges, not their overall distribution which is the same. This creates ground for a comparison that singles out the contribution of the particular assignment of edges, to demonstrate that the way the edges are added has an organizing effect on the network as a whole, with more complex internal structures than would have been produced by assigning the edges randomly. 

The main point is do demonstrate that the cross-links, originally designed for easing navigability between related content have effects in shaping the graph at the mezzo and macro levels. 

Other smaller things:

a. Including triangles and clustering is redundant

Authors' response:

Global clustering is defined as the rate of closed triplets ("triangles") out of all possible triplets. i.e. two nodes connected to the same third node. Potentially, if these are very scarce, the rate of triangles might still be high. Forgoing the clustering coefficient, however, removes some of the context and the insight towards the concept of transitivity in the self-organization of the content, so we decided to keep both. This has now been added to be reflected in the text (table 3).

b. I'm uncertain as to why the terms "content units" and "focal points" are introduced and what purpose they serve. It seems that "content unit" refers to a thread in the forum, and “focal unit” is referring to a node of degree >= 4 but there is no justification as to why that was chosen, nor any theory behind it to suggest the significance of such in knowledge-sharing systems. 

Authors' response:

The term "content unit" refers to a set of a question with its associated answers (if any) and comments (if any). Addressing the entire set of question + answers(+comments) stems from the perception of the entire set as a distinct unit of content within the network. These might be richer in case they contain more answers/ comments, but they stand alone as separate elements regardless. Their inner structure can be hierarchical, up to three layers (question-answers-comments). Amongst them, they can only be connected through a connective element, in this case, a cross-link (but studies have also looked at mutual tags as connectors, for instance (Dankulov et al., 2015)). Ye et al. (Ye et al., 2017) referred to these as knowledge-units. As part of the re-framing of the work, all references to these were altered to knowledge units – first in order to be consistent with existing work on similar environments, and second to reflect each question-answers-comments standalone value. 

The term "focal point" of the network was used to describe a unit that has become relatively connected to other units. Within the entire dataset, hardly any nodes became connected enough to constitute as "hubs", but some nodes began to display potential of acting as local intersections that were at the crossroads of multiple knowledge units. Two indicators were used to identify the emergence of local focal points: 1. The degree of the most connected unit, and 2. The number of units with several connections (4 or above). While the number is somewhat arbitrary, it was designed to capture knowledge units that became connected to several other knowledge units, and so reflect an emergence of hierarchy, units that are more central than others within the collective knowledge product. This may be useful for navigation, or for extracting themes out of the graph by focusing on these nodes. This clarification was added to the manuscript. 

Sometimes the word “sparse” gets used to seemingly describe how disconnected the network is. The ratio presented of links to nodes is actually dense enough to have one connected component and is dense in comparison to many other networks of real systems. In general, however, we measure sparsity and density within-component (so infinite distances get dropped). 

Authors' response:

This has been corrected throughout the manuscript.

Broadly, my recommendation would be to zoom in and reframe the work so that the genuinely good results obtained can be understood through the lens of theory behind knowledge-sharing, instead of trying to develop novel theory about the relationships between network topology and epistemic knowledge-units alongside obtaining results.

Authors' response:

Thank you for this important suggestion. We have tried to incorporate your suggestions and shift the emphasis toward the role of link-sharing within the collaborative process.

---

## [Decision Letter · Decision Letter 3]

23 Feb 2024

Associative linking for collaborative thinking: self-organization of content in online Q&A communities via user-generated links

PONE-D-22-31022R3

Dear Dr. Sher,

We’re pleased to inform you that your manuscript has been judged scientifically suitable for publication and will be formally accepted for publication once it meets all outstanding technical requirements.

Kind regards,

Carlos Henrique Gomes Ferreira, Ph.D.

Academic Editor

PLOS ONE

Reviewers' comments:

Reviewer's Responses to Questions

**Comments to the Author**

1. If the authors have adequately addressed your comments raised in a previous round of review and you feel that this manuscript is now acceptable for publication, you may indicate that here to bypass the “Comments to the Author” section, enter your conflict of interest statement in the “Confidential to Editor” section, and submit your "Accept" recommendation.

Reviewer #1: All comments have been addressed

Reviewer #5: (No Response)

2. Is the manuscript technically sound, and do the data support the conclusions?

Reviewer #1: Yes

Reviewer #5: Yes

3. Has the statistical analysis been performed appropriately and rigorously? 

Reviewer #1: Yes

Reviewer #5: Yes

4. Have the authors made all data underlying the findings in their manuscript fully available?

Reviewer #1: Yes

Reviewer #5: Yes

5. Is the manuscript presented in an intelligible fashion and written in standard English?

Reviewer #1: Yes

Reviewer #5: Yes

6. Review Comments to the Author

Reviewer #1: The authors have addressed all of my previous comments. I am happy with the revised version. I suggest acceptance on Plos One.

Reviewer #5: The manuscript titled "Associative Linking for Collaborative Thinking: Self-Organization of Content in Online Q&A Communities" is an insightful exploration of how virtual collaborative Q&A platforms foster collaborative knowledge creation through dynamic interactions between users and content. This study was motivated by the observation that knowledge within these communities is often fragmented, yet the collective value of this collaboratively formed knowledge base is not fully appreciated. Following research on individual mental semantic networks, the authors investigate the self-organizing nature of knowledge sharing within these communities, as manifested in the associative links created by users.

Using data from eight topic-centered discussions on the Stack Exchange platform, network analysis tools are used to examine the structure of the networks formed by these associative links. By comparing topological indicators of these networks — such as cluster coefficients, degrees of integration and the presence of strongly connected nodes — with those of 1,000 simulated networks generated by random links, the study reveals a striking pattern. The actual networks exhibit a higher degree of clustering, better integration and more strongly connected sites than their randomly generated counterparts, with these differences increasing over time. In addition, the largest connected subgraphs in these networks exhibit modularity, suggesting a nuanced internal organization.

In addition, the study makes the first qualitative observations about how external events that affect content can influence network structures. These findings support the idea that networks formed through associative linking mirror the self-organizing properties of other information networks and highlight the potential of collaborative linking as a mechanism for collective knowledge organization. This research underscores the need to recognize and exploit associative linkage in both theoretical frameworks and practical applications, and provides a compelling argument for its adoption for publication.

I've seen the history of the revisions made in response to the previous rounds in the peer review process.

I agree with the other reviewers that the manuscript has been significantly improved in terms of clarity and organization. The authors have addressed the requested changes, and in my evaluation the manuscript is in a good state to be published.

7. PLOS authors have the option to publish the peer review history of their article (what does this mean?). If published, this will include your full peer review and any attached files.

Reviewer #1: No

Reviewer #5: No

---

## [Editor Report · Acceptance letter]

29 Feb 2024

PONE-D-22-31022R3 

PLOS ONE

Dear Dr. Sher, 

I'm pleased to inform you that your manuscript has been deemed suitable for publication in PLOS ONE. Congratulations! Your manuscript is now being handed over to our production team.

Kind regards, 

on behalf of

Dr. Carlos Henrique Gomes Ferreira 

Academic Editor

PLOS ONE